# Validation of a coupled $\delta^2H_{n\text{-alkane}}$-$\delta^{18}O_{sugar}$ paleohygrometer approach based on a climate chamber experiment

Johannes Hepp[a,b,1,*], Christoph Mayr[c,d], Kazimierz Rozanski[e], Imke Kathrin Schäfer[f], Mario Tuthorn[g,2], Bruno Glaser[b], Dieter Juchelka[g], Willibald Stichler[h], Roland Zech[f,i,3], Michael Zech[b,j,4,#]

[a]Chair of Geomorphology and BayCEER, University of Bayreuth, Universitätsstrasse 30, D-95440 Bayreuth, Germany

[b]Institute of Agricultural and Nutritional Sciences, Soil Biogeochemistry, Martin Luther University Halle-Wittenberg, Von-Seckendorff-Platz 3, D-06120 Halle (Saale), Germany

[c]Institute of Geography, Friedrich-Alexander-University Erlangen-Nürnberg, Wetterkreuz 15, D-91058 Erlangen, Germany

[d]GeoBio-Center & Earth and Environmental Sciences, Ludwig-Maximilian University Munich, Richard-Wagner-Str. 10, D-80333 München, Germany

[e]Faculty of Physics and Applied Computer Science, AGH University of Science and Technology, Al. Mickiewicza 30, PL-30-059 Kraków, Poland

[f]Institute of Geography and Oeschger Centre for Climate Research, University of Bern, Hallerstrasse 12, CH-3012 Bern, Switzerland

[g]Thermo Fisher Scientific, Hanna-Kunath-Str. 11, D-28199 Bremen, Germany

[h]Helmholtz Zentrum München, German Research Center for Environmental Health, Ingolstädter Landstrasse 1, D-85764 Neuherberg, Germany

[i]Institute of Geography, Chair of Physical Geography, Friedrich-Schiller University of Jena, Löbdergraben 32, D-07743 Jena, Germany

[j]Institute of Geography, Heisenberg Chair of Physical Geography with focus on paleoenvironmental research, Technische Universität Dresden, Helmholtzstrasse 10, D-01062 Dresden, Germany

[*]corresponding author: johannes-hepp@gmx.de

---

[1]Present address: Chair of Geomorphology and BayCEER, University of Bayreuth, Universitätsstrasse 30, D-95440 Bayreuth, Germany

[2]Present address: Thermo Fisher Scientific, Hanna-Kunath-Str. 11, D-28199 Bremen, Germany

[3]Present address: Institute of Geography, Chair of Physical Geography, Friedrich-Schiller University of Jena, Löbdergraben 32, D-07743 Jena, Germany

[4]Present address: Institute of Geography, Heisenberg Chair of Physical Geography with focus on paleoenvironmental research, Technische Universität Dresden, Helmholtzstrasse 10, D-01062 Dresden, Germany

## Keywords

hydrogen stable isotopes, oxygen stable isotopes, hemicellulose sugars, leaf waxes, leaf water enrichment, deuterium-excess, relative humidity

## Abstract

The hydrogen isotope composition of leaf wax-derived biomarkers, e.g. long chain $n$-alkanes ($\delta^2H_{n\text{-alkane}}$), is widely applied in paleoclimate. However, a direct reconstruction of the isotope composition of source water based on $\delta^2H_{n\text{-alkane}}$ alone is challenging due to the enrichment of heavy-isotopes during evaporation. The coupling of $\delta^2H_{n\text{-alkane}}$ with $\delta^{18}O$ of hemicellulose-derived sugars ($\delta^{18}O_{sugar}$) has the potential to disentangle this limitation and additionally to allow relative humidity reconstructions. Here, we present $\delta^2H_{n\text{-alkane}}$ as well as $\delta^{18}O_{sugar}$ results obtained from leaves of *Eucalyptus globulus*, *Vicia faba,* and *Brassica oleracea*, which grew under controlled conditions. We addressed the following questions (i) do $\delta^2H_{n\text{-alkane}}$ and $\delta^{18}O_{sugar}$ values allow reconstructions of leaf water isotope composition, (ii) how accurately does the reconstructed leaf-water-isotope composition enable relative humidity (RH) reconstruction, and (iii) does the coupling of $\delta^2H_{n\text{-alkane}}$ and $\delta^{18}O_{sugar}$ enable a robust source water calculation?

For all investigated species, the $n$-alkane $n$-$C_{29}$ was most abundant and therefore used for compound-specific $\delta^2H$ measurements. For *Vicia faba*, additionally the $\delta^2H$ values of $n$-$C_{31}$ could be evaluated robustly. Regarding hemicellulose-derived monosaccharides, arabinose and xylose were most abundant and their $\delta^{18}O$ values were therefore used to calculate weighted mean leaf $\delta^{18}O_{sugar}$ values. Both $\delta^2H_{n\text{-alkane}}$ and $\delta^{18}O_{sugar}$ yielded significant correlations with $\delta^2H_{leaf\text{-water}}$ and $\delta^{18}O_{leaf\text{-water}}$, respectively ($r^2 = 0.45$ and $0.85$, respectively; $p < 0.001$, n = 24). Mean fractionation factors between biomarkers and leaf water were found to be -156‰ (ranging from -133 to -192‰) for $\varepsilon_{n\text{-alkane/leaf-water}}$ and +27.3‰ (ranging from +23.0 to 32.3‰) for $\varepsilon_{sugar/leaf\text{-water}}$, respectively. Modelled $RH_{air}$ values from a Craig-Gordon model using measured $T_{air}$, $\delta^2H_{leaf\text{-water}}$ and $\delta^{18}O_{leaf\text{-water}}$ as input correlate highly significantly with modeled $RH_{air}$ values ($R^2 = 0.84$, $p < 0.001$, RMSE = 6%). When coupling $\delta^2H_{n\text{-alkane}}$ and $\delta^{18}O_{sugar}$ values, the correlation of modeled $RH_{air}$ values with measured $RH_{air}$ values is weaker but still highly significant with $R^2 = 0.54$ ($p < 0.001$, RMSE = 10%). Finally, the reconstructed source water isotope composition ($\delta^2H_s$ and $\delta^{18}O_s$) as calculated from our coupled approach matches the source water in the climate chamber experiment ($\delta^2H_{tank\text{-water}}$ and $\delta^{18}O_{tank\text{-water}}$). This highlights the great potential of the coupled $\delta^2H_{n\text{-alkane}}$-$\delta^{18}O_{sugar}$ paleohygrometer approach for paleoclimate and relative humidity reconstructions.

## 1 Introduction

Leaf wax-derived biomarkers such as long chain *n*-alkanes and their stable hydrogen isotope composition ($\delta^2H_{n\text{-alkane}}$) are widely applied in paleoclimatology. Sedimentary $\delta^2H_{n\text{-alkane}}$ values correlate with $\delta^2H$ of precipitation (Huang et al., 2004; Mügler et al., 2008; Sachse et al., 2004; Sauer et al., 2001),
confirming the high potential of $\delta^2H_{n\text{-alkane}}$ to establish $\delta^2H$ records of past precipitation (Hou et al., 2008; Rao et al., 2009; Sachse et al., 2012). However, this interpretation is challenged by heavy isotope enrichment during evapotranspiration (e.g. Zech et al., 2015). Apart from studies of sedimentary cellulose (Heyng et al., 2014; Wissel et al., 2008), the oxygen stable isotope composition of sugar biomarkers ($\delta^{18}O_{sugar}$) emerged as complementary paleoclimate proxy during the last decade (Hepp et
al., 2015, 2017, Zech et al., 2013a, 2014a; Zech and Glaser, 2009). The interpretation of the $\delta^{18}O_{sugar}$ values is comparable to those of $\delta^2H_{n\text{-alkane}}$. When sugars originate primarily from leaf biomass of higher terrestrial plants, they reflect the plant source water (which is often directly linked to the local precipitation) modified by evapotranspirative enrichment of the leaf water (Tuthorn et al., 2014; Zech et al., 2014a). The coupling of $\delta^2H_{n\text{-alkane}}$ with $\delta^{18}O_{sugar}$ values allows quantifying the leaf-water isotope
enrichment and relative air humidity (Zech et al., 2013a). This approach was validated by Tuthorn et al. (2015) by applying it to topsoil samples along a climate transect in Argentina. Accordingly, the biomarker-derived relative air humidity values correlate significantly with actual relative air humidity, highlighting the potential of the $\delta^2H_{n\text{-alkane}}$-$\delta^{18}O_{sugar}$ paleohygrometer approach.

The coupled approach is based on the observation that the isotope signature of precipitation ($\delta^2H_{precipitation}$ and $\delta^{18}O_{precipitation}$) typically plots near to the global meteoric water line (GMWL) in a two-dimensional $\delta^2H$-$\delta^{18}O$ diagram. The GMWL is characterized by the equation $\delta^2H_{precipitation} = 8 \cdot \delta^{18}O_{precipitation} + 10$ (Craig, 1961; Dansgaard, 1964). In many cases, the local precipitation is directly linked to the source water of plants, which is indeed soil water and shallow groundwater. The isotope
composition of xylem water readily reflects these sources (e.g. Dawson, 1993). However, leaf-derived biomarkers reflect the leaf water isotope composition, which is, unlike xylem water, prone to evapotranspiration (e.g. Barbour and Farquhar, 2000; Helliker and Ehleringer, 2002; Cernusak et al., 2003; Barbour et al., 2004; Cernusak et al., 2005; Feakins and Sessions, 2010; Kahmen et al., 2011; Sachse et al., 2012; Kahmen, et al., 2013; Tipple et al., 2013; Lehmann et al., 2017; Liu et al., 2017).
During daytime, the leaf water is typically enriched in the heavy isotope compared to the source water because of evapotranspiration through stomata. Thereby, lighter water isotopes evaporate preferentially, which leads to gradual enrichment of heavier isotopes compared to precipitation. The degree of enrichment by evapotranspiration is mainly controlled by the relative air humidity ($RH_{air}$) in the direct surrounding of the plant leaves (e.g. Cernusak et al., 2016). Although the biomarkers reflect
the isotope composition of leaf water, there is still a modification of its isotope signature by fractionation during biosynthesis, leading to an offset between leaf water and biomarker isotope compositions. In case the biosynthetic fractionation is known and constant, $RH_{air}$ can be calculated from coupling $\delta^2H_{n\text{-alkane}}$ with $\delta^{18}O_{sugar}$ values.

First applications of this approach for paleoclimatic reconstruction (Hepp et al., 2017, 2019; Zech et al., 2013a) and climate transect validation studies (Hepp et al., 2020; Lemma et al., 2021; Strobel et al., 2020; Tuthorn et al., 2015) revealed promising results. Furthermore, the stability of the biomarker isotope signals during degradation was studied (Zech et al., 2011, 2012). In brief, *n*-alkanes and sugars can be extracted compound-specifically from plants, soils and a wide range of different sediments
retaining the isotope signal of intact plant tissues.

The overall aim of this study is to evaluate the $\delta^2H_{n\text{-alkane}}$-$\delta^{18}O_{sugar}$ paleohygrometer approach by applying it to plant leaf material from three different plants grown in a climate chamber experiment under controlled conditions. More specifically, we addressed the following questions:

(i)    which *n*-alkanes and monosaccharides can be used to obtain $\delta^2H_{n\text{-alkane}}$ and $\delta^{18}O_{sugar}$ results of the leaves grown in our climate chamber experiment, respectively,

(ii)    how precisely do $\delta^2H_{n\text{-alkane}}$ and $\delta^{18}O_{sugar}$ values allow reconstructing $\delta^2H_{leaf\text{-water}}$ and $\delta^{18}O_{leaf\text{-water}}$, respectively,

(iii)    how accurately does the leaf water isotope composition reflect $RH_{air}$,

(iv)    does the coupling of $\delta^2H_{n\text{-alkane}}$ and $\delta^{18}O_{sugar}$ enable a $RH_{air}$ reconstruction, and

(v)    how robust are source water calculations?

## 2 Material and Methods

### 2.1 Climate chamber experiment

A phytotron experiment was conducted at the Helmholtz Zentrum München in Neuherberg during winter 2000/2001 (Mayr, 2002). Three different dicotyledon plant species (*Eucalyptus globulus*, *Vicia faba* var. *minor* and *Brassica oleracea* var. *medullosa*) were grown in eight chambers for 56 days under seven distinct climatic conditions (same conditions in chambers 4 and 8). All three species belong to $C_3$ plants and were primarily chosen because they are different in terms of morphology and physiology.

Two of the species are herbaceous (*Vicia faba* var. *minor* and *Brassica oleracea* var. *medullosa*), while *Eucalyptus globulus* is a lignifying tree. Nevertheless, all three species form a stem. The different habitus of the three plants allows to check for inter-species differences in isotope fractionation (Mayr, 2002). Similar isotope patterns in the three plants could allow generalizing our results for a wide range of $C_3$ plants, independent of their habitus. An additional, equally important criterion for the choice of

the plants was the resilience of the taxa to the experimental climatic conditions and sufficient growth during the experiments. Air temperature ($T_{air}$) was set to 14, 18, 24 and 30°C and $RH_{air}$ to around 20, 30, 50, and 70% during the daytime of constant climate conditions (between 11 a.m. and 4 p.m.) (Fig. 1A). During the rest of the day diurnal variations typical for natural conditions were aimed for (for more details see Mayr, 2002).

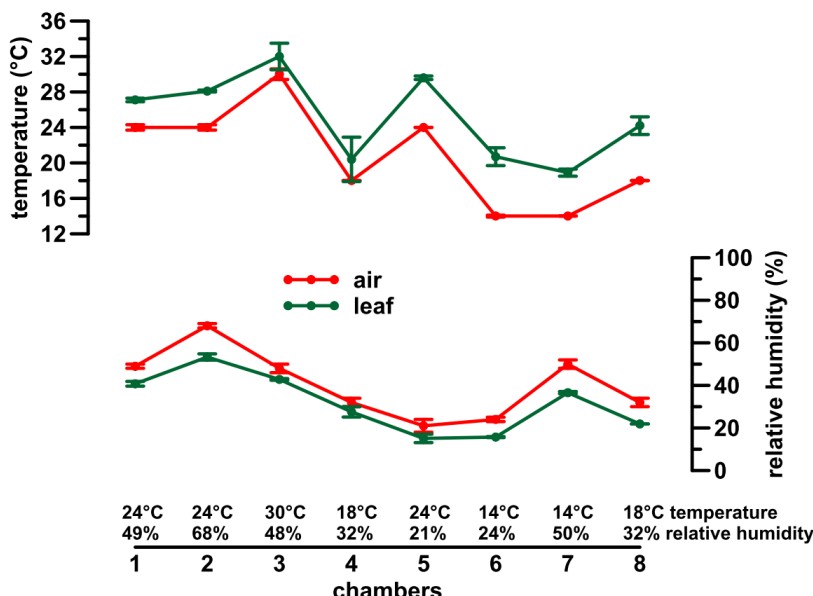

**Fig. 1:** A: Climate chamber conditions. Error bars represent analytical standard deviation of the respective measurements. For more details see section 2.2 and Mayr (2002).

The chambers had a high fresh air supply rate (750 m³ h$^{-1}$), avoiding therefore an evaporative isotope enrichment in the chambers. Furthermore, uniform irrigation conditions were guaranteed via an automatic irrigation system, which was controlled by tensiometers installed in 9 cm substrate depth. The tank water used for irrigation was sampled periodically (intervals of one to three days) over the whole experiment and revealed only minor variability in its isotope composition ($\delta^{18}O_{tank-water}$ = -10.7 ± 0.3‰ standard deviation (σ); $\delta^{2}H_{tank-water}$ = -7 ± 1‰ σ). Once a week, soil water (via ceramic cups in 13 cm soil depth) and atmospheric water vapor (via dry ice condensation traps) was sampled ($\delta^{2}H_{soil-water}$, $\delta^{18}O_{soil-water}$ and $\delta^{2}H_{atmospheric-water-vapor}$, $\delta^{18}O_{atmospheric-water-vapor}$). Additionally, leaf temperature ($T_{leaf}$) was calculated from gas exchange measurements, at least once a week (Mayr, 2002).

In order to analyze stable hydrogen and oxygen isotope composition of leaf ($\delta^{2}H_{leaf-water}$, $\delta^{18}O_{leaf-water}$) and stem water, the plants were harvested at the end of the experiment. The vacuum distillation method was used for the extraction of the plant water. It should be noted that stem water is a mixture between phloem and xylem water. Only the latter reflects the isotope composition of soil water. For simplification, stem water is referred to as xylem water in the following ($\delta^{2}H_{xylem-water}$, $\delta^{18}O_{xylem-water}$).

For more details about the experiment, the (plant) water isotope results, and gas exchange measurements we refer to the original publication (Mayr, 2002).

## 2.2 Leaf biomarker extraction and compound-specific stable isotope analysis

A total of 24 leaf samples were prepared according to Schäfer et al. (2016) for compound specific $\delta^{2}H$ measurements of *n*-alkanes, at the Institute of Geography, Group of Biogeochemistry and Paleoclimate, University of Bern. Microwave extraction with 15 ml dichloromethane (DCM)/methanol (MeOH) 9:1 (v:v) at 100°C for 1 h was conducted. The resulting total lipid extracts were purified and separated using aminopropyl-silica-gel (Supelco, 45 μm) pipette columns. The hydrocarbon fractions (containing *n*-alkanes) were eluted with *n*-hexane and cleaned via silver nitrate-coated silica gel pipettes (Supelco, 60-200 mesh) and zeolite (Geokleen Ltd.) columns. The $\delta^{2}H$ measurements of the dominant *n*-alkanes (*n*-C$_{29}$ and *n*-C$_{31}$) were performed on a GC-$^{2}$H-pyrolysis-IRMS system, consisting of an Agilent 7890A gas chromatograph (GC) coupled to an IsoPrime 100 isotope-ratio-mass spectrometer (IRMS) via a GC5 pyrolysis/combustion interface operating in pyrolysis mode with a Cr (ChromeHD) reactor at 1000 °C. The compound-specific $\delta^{2}H$ values were calibrated against a standard alkane mix (*n*-C$_{27}$, *n*-C$_{29}$, *n*-C$_{33}$) with known isotope composition (A. Schimmelmann, University of Indiana), measured twice every six sample injections. Standard deviation of the triplicate measurements was typically ≤5‰. The H$^{3+}$ factor stayed constant during the measurements.

Additionally, the leaf samples were dried and finely ground in preparation for $\delta^{18}O$ analysis of hemicellulose-derived sugars (modified from Zech and Glaser, 2009) at the Institute of Agricultural and Nutritional Sciences, Soil Biogeochemistry, Martin Luther University Halle-Wittenberg. The hemicellulose sugars were hydrolytically extracted for 4 h at 105 °C using 4M trifluoroacetic acid (Amelung et al., 1996) and purified via XAD-7 and Dowex 50WX8 columns. Prior to the methylboronic-acid (MBA) derivatization (4 mg of MBA in 400 μl dry pyridine for 1 h at 60°C), the cleaned sugars were frozen and freeze-dried overnight (Gross and Glaser, 2004; Knapp, 1979). Compound-specific $\delta^{18}O$ measurements were performed on a Trace GC 2000 coupled to a Delta V Advantage IRMS via an $^{18}O$-pyrolysis reactor (GC IsoLink) and a ConFlo IV interface (all devices from Thermo Fisher Scientific, Bremen, Germany). The sample batches were measured along with embedded co-derivatized standard batches, which contained arabinose, fucose, xylose, and rhamnose in different concentrations of known $\delta^{18}O$ value. The $\delta^{18}O$ values of the standard sugars were determined via temperature conversion/elemental analysis-IRMS coupling at the Institute of Plant Sciences, ETH Zurich, Switzerland

(Zech and Glaser, 2009). This procedure allows corrections for possible amount dependencies (Zech and Glaser, 2009) and ensures the "Principle of Identical Treatment" (Werner and Brand, 2001). Standard deviations for the triplicate measurements were 0.9‰ and 2.2‰ (average over all investigated samples) for arabinose and xylose, respectively. We focus on arabinose and xylose in this study because they were (i) the dominant peaks in all chromatograms, and (ii) previously found to
strongly predominate over fucose (and rhamnose) in terrestrial plants and soils (Hepp et al., 2016).

All δ values are expressed in ‰ as isotope ratios (R = $^{18}O/^{16}O$ or $^{2}H/^{1}H$) relative to the Vienna Standard Mean Ocean Water (VSMOW) standard in the common delta notation (δ = $(R_{sample} - R_{standard})/R_{standard}$; e.g. Coplen, 2011).

## 2.3 Framework for coupling δ²H$_{n\text{-alkane}}$ with δ¹⁸O$_{sugar}$ results

### 2.3.1 Deuterium-excess of leaf water and relative humidity

The coupled approach is based on the observation that isotope composition of global precipitation plots typically close to the GMWL (δ²H$_{precipitation}$ = 8 · δ¹⁸O$_{precipitation}$ + 10‰; Craig, 1961; Fig. 4). The soil
water and shallow groundwater, which acts as source water for plants, can often directly be related to the local precipitation. However, especially during daytime, leaf water is typically enriched in heavy isotopes compared to the precipitation due to evapotranspiration through the stomata, therefore plotting to the right of the GMWL (Fig. 4; e.g. Allison et al., 1985; Bariac et al., 1994; Walker and Brunel, 1990). Under natural climatic conditions, the leaf water reservoir at the evaporative sites is frequently
assumed to be in isotope steady-state (Allison et al., 1985; Bariac et al., 1994; Gat et al., 2007; Walker and Brunel, 1990), meaning that isotope composition of transpired water vapour is equal to the isotope composition of the source water utilized by plants during the evapotranspiration process. The Craig-Gordon model (e.g. Flanagan et al., 1991; Roden and Ehleringer, 1999) approximates the isotope processes in leaf water in δ terms (e.g. Barbour et al., 2004):

$$\delta_e \approx \delta_s + \varepsilon^* + \varepsilon_k + (\delta_a - \delta_s - \varepsilon_k)\, \frac{e_a}{e_i}, \qquad \text{(Equation 1)}$$

where $\delta_e$, $\delta_s$ and $\delta_a$ are the hydrogen and oxygen isotope compositions of leaf water at the evaporative sites, source water and atmospheric water vapor, respectively. The equilibrium enrichment ($\varepsilon^*$) is expressed as $(1-1/\alpha_{L/V}) \cdot 10^3$, where $\alpha_{L/V}$ is the equilibrium fractionation between liquid and vapor in ‰. The kinetic fractionation parameter ($\varepsilon_k$) describes the water vapor diffusion from intracellular air space through the stomata and the boundary layer into to the atmosphere, and $e_a/e_i$ is the ratio of the
atmospheric to intracellular vapor pressure.

In a δ²H-δ¹⁸O diagram, the isotope composition of the leaf water as well as the source water can be described as deuterium-excess (d) values by using the equation of Dansgaard (1964), with d = δ²H - 8 · δ¹⁸O. This allows rewriting Eq. 1, in which hydrogen and oxygen isotopes have to be handled in separate equations, in one equation:

$$d_e \approx d_s + \left(\varepsilon_2^* - 8 \cdot \varepsilon_{18}^*\right) + \left(C_k^2 - 8 \cdot C_k^{18}\right) + \left[d_a - d_s - \left(C_k^2 - 8 \cdot C_k^{18}\right)\right] \cdot \frac{e_a}{e_i}, \qquad \text{(Equation 2)}$$

where $d_e$, $d_s$ and $d_a$ are the deuterium-excess values of leaf water at evaporative sites, source water and atmospheric water vapor, respectively. The kinetic fractionation ($\varepsilon_k$) is typically related to stomata and boundary layer resistances to water flux (Farquhar et al., 1989). We used the kinetic enrichment factor ($C_k$) instead of $\varepsilon_k$ to be close to paleo studies were direct measurements of plant physiology is not possible. The kinetic enrichment factor is derived from a more generalized form of the Craig-
Gordon model for describing the kinetic isotope enrichment for $^{2}H$ and $^{18}O$ ($C_k^2$ and $C_k^{18}$, respectively) (Craig and Gordon, 1965; Gat and Bowser, 1991). If the plant source water and the local atmospheric

water vapor are in isotope equilibrium, the term $\delta_a$ - $\delta_s$ in Eq. 1 can be approximated by -ε*. Thus, Eq. 2 can be reduced to:

$$d_e \approx d_s + \left( \varepsilon_2^* - 8 \cdot \varepsilon_{18}^* + C_k^2 - 8 \cdot C_k^{18} \right) \cdot \left( 1 - \frac{e_a}{e_i} \right). \tag{Equation 3}$$

The actual atmospheric vapor pressure ($e_a$) and the leaf vapor pressure ($e_i$) in kPa can be derived from Eqs. 4 and 5 by using $T_{air}$ (Buck, 1981):

$$e_a = 0.61121 \cdot e^{[17.502 \cdot T_{air} / (T_{air} + 240.97)]} \cdot RH_{air} \tag{Equation 4}$$

$$e_i = 0.61121 \cdot e^{[17.502 \cdot T_{air} / (T_{air} + 240.97)]}. \tag{Equation 5}$$

When $e_i$ is calculated as in Eq. 5, $e_a/e_i$ represents $RH_{air}$ (ranging between 0 and 1, representing 0 to 100% relative humidity). We are aware, that the Craig-Gordon model would require $T_{leaf}$ values for calculating $e_i$ values. However, the RH reconstruction methodological framework presented is attempted to paleo studies for which the $T_{leaf}$ parameter is probably rather difficult to achieve. However, as can be seen in Fig. 1, leaf temperature is very close to air temperature.

With rearranging Eq. 3, an equation is given to derive relative humidity values (Eq. 6):

$$RH_{air} \approx 1 - \frac{d_e - d_s}{\left( \varepsilon_2^* - 8 \cdot \varepsilon_{18}^* + C_k^2 - 8 \cdot C_k^{18} \right)}. \tag{Equation 6}$$

Equilibrium fractionation parameters ($\varepsilon_2^*$ and $\varepsilon_{18}^*$) can be calculated from empirical equations of Horita and Wesolowski (1994) by using the climate chamber $T_{air}$ values. The kinetic fractionation parameters ($C_k^2$ and $C_k^{18}$) for $^2H$ and $^{18}O$, respectively, are set to 25.1 and 28.5‰ according to Merlivat (1978), who reported maximum values during the molecular diffusion process of water through a stagnant boundary layer. When using supplementary data of Cernusak et al., (2016), $\varepsilon_k$ values of broadleaf trees and shrubs over broad climatic conditions can be calculated which are well in the range with the used $C_k^2$ and $C_k^{18}$ values (23.9 ± 0.9 and 26.7‰ ± 1.0 for $\varepsilon_k^2$ and $\varepsilon_k^{18}$, respectively).

If $\delta^2H_{leaf-water}$ and $\delta^{18}O_{leaf-water}$ can be reconstructed from the measured δ values of *n*-alkane and sugar biomarkers, this framework provides a powerful tool to establish relative humidity records from sedimentary archives (Hepp et al., 2017; Zech et al., 2013a).

To reconstruct the isotope composition of leaf water it is assumed that fractionation factors of −160‰ for $^2H$ of alkanes *n*-$C_{29}$ and *n*-$C_{31}$ ($\varepsilon^2_{bio}$; Sachse et al., 2012; Sessions et al., 1999), and +27‰ for $^{18}O$ of the hemicellulose-derived sugars arabinose and xylose ($\varepsilon^{18}_{bio}$; Cernusak et al., 2003; Schmidt et al., 2001; Sternberg et al., 1986; Yakir and DeNiro, 1990) can be applied:

$$\text{alkane-based } \delta^2H_{leaf-water} = (\delta^2H_{n\text{-alkane}} − \varepsilon^2_{bio})/(1 + \varepsilon^2_{bio}/1000) \tag{Equation 7}$$

$$\text{sugar-based } \delta^{18}O_{leaf-water} = (\delta^{18}O_{sugar} − \varepsilon^{18}_{bio})/(1 + \varepsilon^{18}_{bio}/1000). \tag{Equation 8}$$

### 2.3.2 Isotope composition of plant source water

In a $\delta^2H$-$\delta^{18}O$ diagram, the hydrogen and oxygen isotope composition of the plant source water ($\delta^2H_s$ and $\delta^{18}O_s$, respectively) can be reconstructed via the slope of the individual leaf water evapotranspiration lines (LEL´s; Craig and Gordon, 1965; Gat and Bowser, 1991). The LEL slope ($S_{LEL}$) can be derived from Eq. 9:

$$S_{LEL} \approx \frac{\varepsilon_2^* + C_k^2 \cdot \left(1 - \frac{e_a}{e_i}\right)}{\varepsilon_{18}^* + C_k^{18} \cdot \left(1 - \frac{e_a}{e_i}\right)} \approx \frac{\varepsilon_2^* + C_k^2}{\varepsilon_{18}^* + C_k^{18}},$$ (Equation 9)

where all variables are defined as in section 2.3.1. The $\delta^2 H_s$ and $\delta^{18}O_s$ values can then be calculated for
each leaf water data point via the intersect between the individual LEL´s with the GMWL. The $\delta^2 H_s$ and
$\delta^{18}O_s$ model results can then be compared to the measured $\delta^2 H_{tank\text{-}water}$ and $\delta^{18}O_{tank\text{-}water}$ values.

### 2.4 Modeling and isotope fractionation calculations

The $d_e$ values are modeled using Eq. 3 and measured $RH_{air}$ as input, which can be compared to the
deuterium-excess via the measured $\delta^2 H_{leaf\text{-}water}$ and $\delta^{18}O_{leaf\text{-}water}$ values. The $RH_{air}$ can be derived from
Eq. 6 and compared to the measured ones. In a next step, reconstructed (biomarker-based) deuterium-
excess$_{leaf\text{-}water}$ was used as input for Eq. 6 and compared to the measured $RH_{air}$ values. This model
represents a simplified approach because $\delta_a - \delta_s$ are approximated by $-\varepsilon^*$ (see section 2.3). In all
equations where $\delta_s$ and $d_s$ are needed as input the measured $\delta^2 H_{tank\text{-}water}$ and $\delta^{18}O_{tank\text{-}water}$ were used
for calculations. The equilibrium and kinetic fractionation factors were set as described in section 2.3.
Model quality was overall assessed by calculating the coefficient of determination $\left[R^2 = 1 - \sum (\text{modeled - measured})^2 / \sum (\text{measured - measured mean})^2\right]$ and the root mean square error
$\left[RMSE = \sqrt{\left(\frac{1}{n} \cdot \sum (\text{modeled - measured})^2\right)}\right]$. The $R^2$ is not equal to the $r^2$, which provides here the
fraction of variance explained by a linear regression between a dependent (y) and an explanatory
variable $[r^2 = 1 - \sum (y - \text{fitted y})^2 / \sum (y - \text{mean y})^2]$ (R Core Team, 2015).

The fractionation between the measured leaf biomarkers and leaf water can be described by the
following equations (e.g. Coplen, 2011):

$\varepsilon_{n\text{-}alkane/leaf\text{-}water} = (\delta^2 H_{n\text{-}alkane} - \delta^2 H_{leaf\text{-}water}) / (1 + \delta^2 H_{leaf\text{-}water}/1000)$ (Equation 10)

$\varepsilon_{sugar/leaf\text{-}water} = (\delta^{18}O_{sugar} + \delta^{18}O_{leaf\text{-}water}) / (1 + \delta^{18}O_{leaf\text{-}water}/1000).$ (Equation 11)

In order to provide a 1 σ range bracketing the modeled results and calculations, they were additionally
run with values generated by subtracting/adding the individual σ to the average.

All calculations and statistical analysis were realized in R (version 3.2.2; R Core Team, 2015).

## 3 Results and Discussion

### 3.1 Compound-specific isotope results

#### 3.1.1 Leaf wax-derived *n*-alkanes

The investigated leaf material shows a dominance of the $n$-$C_{29}$ alkane homologue. Such a dominance of $n$-$C_{29}$ in *Brassica oleracea* and *Eucalyptus globulus* was also reported by Ali et al. (2005) and Herbin and Robins (1968). *Vicia faba* leaf samples additionally revealed a high abundance of $n$-$C_{31}$. This agrees with results from Maffei (1996) and enables a robust determination of compound-specific $\delta^2H$ values for $C_{29}$ and $C_{31}$, respectively. The $\delta^2H_{n\text{-alkane}}$ values of *Vicia faba* are therefore calculated as weighted mean. Figure 2 illustrates the $\delta^2H_{n\text{-alkane}}$ results along with isotope data for leaf, xylem and soil water (the latter were originally published in Mayr 2002). The $\delta^2H_{n\text{-alkane}}$ values ranged from -213 to -144‰ across all three plant species. As revealed by overlapping notches in the respective box plots, there is no statistically significant difference of the median between the three species (Fig. A1A; McGill et al., 1978). Figures A1A and 2, moreover show that the range of $\delta^2H_{n\text{-alkane}}$ values is largest for *Eucalyptus globulus*. However, the low number of samples per plant species prohibits a robust interpretation.

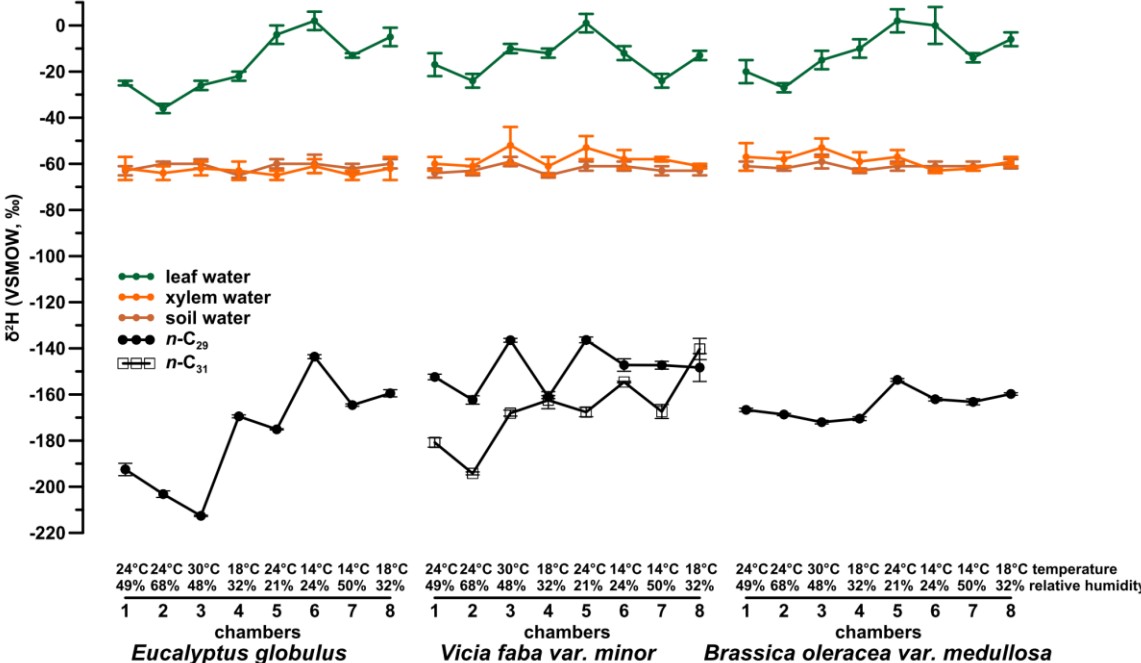

**Fig. 2:** $\delta^2H$ values of plant water (leaf water, xylem water and soil water) isotope compositions (in green, orange and brown, respectively) and the isotope composition of the investigated leaf wax $n$-alkanes $n$-$C_{29}$ and $n$-$C_{31}$ as dots and open squares, respectively.

#### 3.1.2 Hemicellulose-derived sugars

The investigated leaf samples yielded substantially higher amounts of arabinose and xylose compared to fucose and rhamnose. This is in agreement with sugar patterns reported for higher plants (D'Souza et al., 2005; Hepp et al., 2016; Jia et al., 2008; Prietzel et al., 2013; Zech et al., 2012, 2014a) and hampers a robust data evaluation of fucose and rhamnose. Therefore, the $\delta^{18}O$ values were investigated for the pentoses arabinose and xylose and range from 30 to 47‰ and 30 to 50‰ (Fig. 3), respectively. Additionally, the isotope data for leaf, xylem and soil water are shown in the figure (originally published in Mayr 2002). No considerable difference in the $\delta^{18}O$ values of arabinose and xylose can be seen in the $\delta^{18}O$ data ($r^2 = 0.7$, $p < 0.001$, $n = 24$). This is in line with findings of Zech and Glaser (2009), Zech et al. (2012), Zech et al. (2013b) and Zech et al. (2014b) but contradicting with

315 slightly more positive $\delta^{18}O_{arabinose}$ values compared to $\delta^{18}O_{xylose}$ values reported by Zech et al. (2013a) and Tuthorn et al. (2014). Zech et al. (2013a) and Tuthorn et al. (2014) argue that a biosynthetic fractionation could be the reason for that difference. This is based on the fact that arabinose is biosynthesized via an epimerase from xylose (Altermatt and Neish, 1956; Burget et al., 2003; Harper and Bar-Peled, 2002). Nevertheless, the $\delta^{18}O$ values of arabinose and xylose were here combined as

weighted mean (as $\delta^{18}O_{sugar}$ values) for further data interpretation. Overall, the $\delta^{18}O_{sugar}$ values are not significantly different between the three investigated plant species (Fig. A1B).

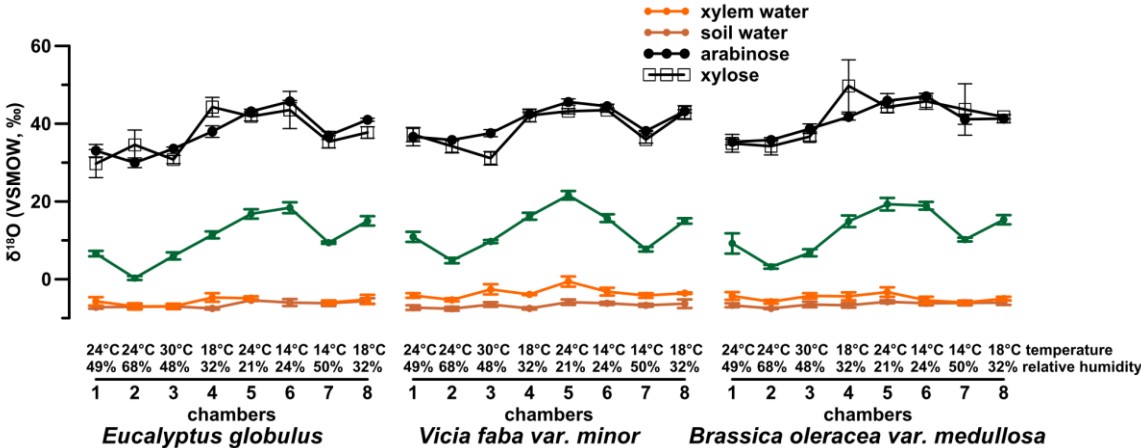

**Fig. 3:** $\delta$18O values of plant water (leaf water, xylem water and soil water) isotope compositions (in green, orange and brown, respectively) and the isotope composition of the investigated hemicellulose-

325 derived sugars: arabinose and xylose as dots and open squares, respectively.

### 3.1.3 $\delta^2H$-$\delta^{18}O$ diagram

A comparison of compound-specific isotopes of leaf hemicellulose-derived sugars and leaf wax-derived *n*-alkanes with leaf, xylem, soil and tank water (compare Figs. 2, 3 and Fig. 4) reveals that soil and xylem

water plot close to the tank water. Only a slight heavy-isotope enrichment in the soil and xylem water was observed (c.f. the offset between soil and xylem water compared to the tank water). A larger evaporation effect of these water pools was inhibited by the high fresh air supply to the climate chambers (see section 2.1). Thus, the leaf water shows a clear heavy-isotope enrichment due to evapotranspiration. This enrichment strongly differs between the climate chambers, depending mainly

on T and RH conditions. The biomarker results furthermore follow the leaf water with a certain offset, which is $\varepsilon_{bio}$.

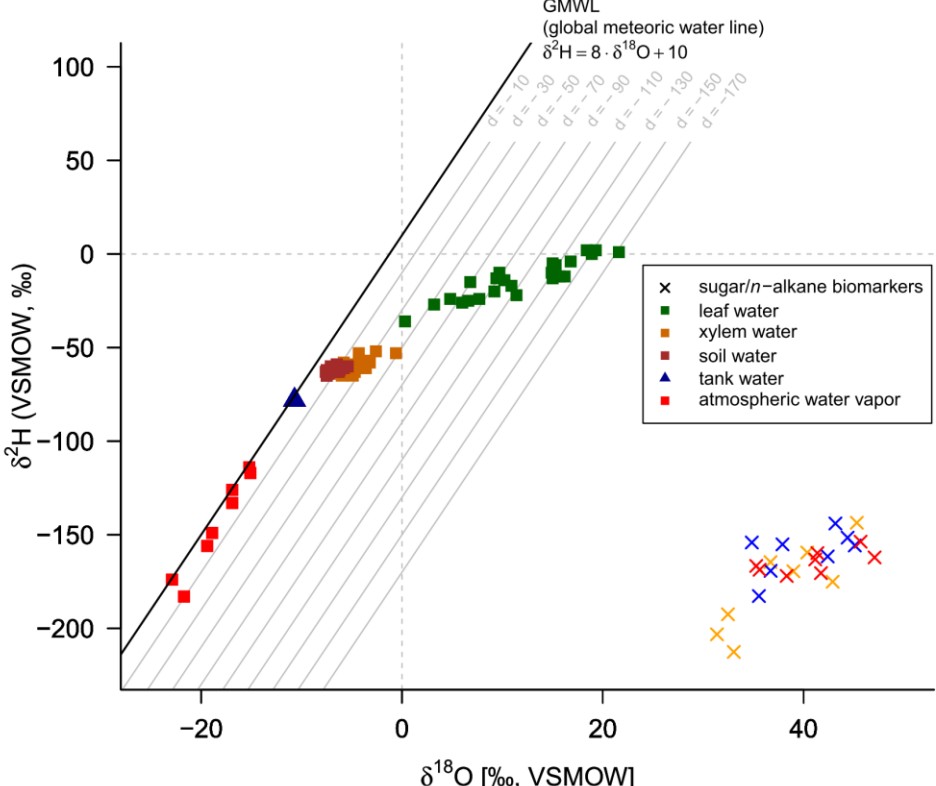

**Fig. 4:** $\delta^2$H-$\delta^{18}$O diagram illustrating the isotope composition of the biomarkers: $\delta^2$H values of the leaf wax *n*-alkanes ($C_{29}$ for *Eucalyptus globulus* and *Brassica oleracea* and weighted mean of $C_{29}$ and $C_{31}$ for *Vicia faba*) and $\delta^{18}$O values of the hemicellulose-derived sugars arabinose and xylose (red crosses = *Brassica oleracea*, orange crosses = *Eucalyptus globulus*, blue crosses = *Vicia faba*). The measured isotope compositions of leaf water (green squares), xylem water (orange squares), soil water (brown squares), atmospheric water vapor (red squares) and the tank water used for irrigation (blue triangle), which plot very close to the global meteoric water line (GMWL). The deuterium-excess with respect to the GMWL is marked as d in ‰.

## 3.2 Do *n*-alkane and sugar biomarkers reflect the isotope composition of leaf water?

### 3.2.1 $\delta^2$H$_{n\text{-alkane}}$ vs. $\delta^2$H$_{\text{leaf-water}}$

The $\delta^2$H$_{n\text{-alkane}}$ dataset including all plant species reveals a significant correlation with $\delta^2$H$_{\text{leaf-water}}$ ($r^2$ = 0.45 with $p < 0.001$) (Fig. 5A). A slope of 1.1 and an intercept of -152‰ characterize the linear relationship. Since it is well known that measured leaf water is not always equal to the specific water pool in which the *n*-alkanes were biosynthesized (e.g. Tipple et al., 2015), this could explain the observed rather low $r^2$ value. The correlation between the $\delta^2$H$_{n\text{-alkane}}$ and $\delta^2$H$_{\text{leaf-water}}$ presented here is still well in range with the literature. Feakins and Sessions (2010) presented *n*-alkane ($C_{29}$ and $C_{31}$) and leaf water $\delta^2$H data from typical plant species (excluding grasses) along a southern California aridity gradient, revealing that only $\delta^2$H of *n*-$C_{29}$ is significantly correlated with leaf water ($r^2$ = 0.24, $p < 0.1$, n = 16; based on the associated supplementary data). Another field dataset from the temperate forest at Brown's Lake Bog, Ohio, USA revealed significant correlations between $\delta^2$H of *n*-$C_{29}$ or *n*-$C_{31}$ and leaf water of *Prunus serotina*, *Acer saccharinum*, *Quercus rubra*, *Quercus alba,* and *Ulmus americana* ($r^2$ = 0.49, $p < 0.001$, n = 38; $r^2$ = 0.59, $p < 0.001$, n = 29; as derived form the supplement material of Freimuth et al., 2017). Data from a controlled climate chamber experiment using two tree species show a highly significant relationship between $\delta^2$H of leaf wax *n*-alkanes and leaf water (with $C_{31}$ of *Betula*

*occidentalis* and $C_{29}$ of *Populus fremontii*; $r^2 = 0.96$, p < 0.001, n = 24; derived from supplementary data of Tipple et al., 2015).

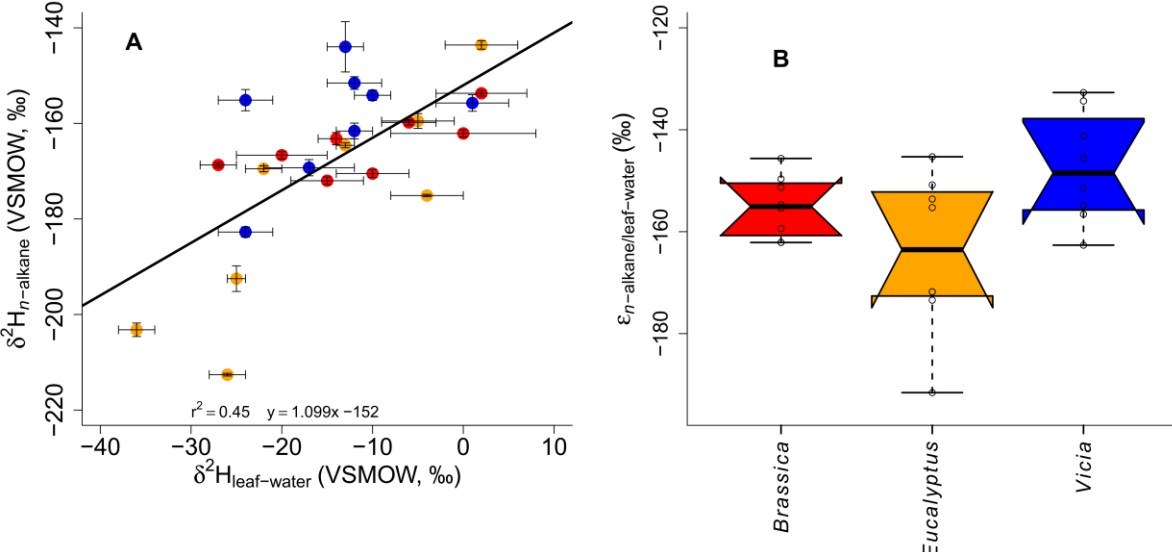

**Fig. 5:** (A) Scatterplot depicting the relationship between $\delta^2H_{n\text{-alkane}}$ and $\delta^2H_{leaf\text{-water}}$. Error bars of the δ values represent standard deviation of repeated measurements (see section 2.2 and Mayr, 2002). (B) Boxplot comprising the plant-specific fractionation $\varepsilon_{n\text{-alkane/leaf-water}}$ according Eq. 10, showing median (thick black line), interquartile range (IQR) with upper (75%) and lower (25%) quartiles, lower and upper whiskers, which are restricted to 1.5 · IQR. Outside the 1.5 · IQR space, the data points are marked with a dot. The notches are extend to ± 1.58 ·IQR/√n, by convention and give a 95% confidence interval for the difference of two medians (McGill et al., 1978). *Brassica oleracea, Eucalyptus globulus* and *Vicia faba* samples are shown in red, orange and blue, respectively.

It seems that each plant type shows a different $\delta^2H_{n\text{-alkane}}$ to $\delta^2H_{leaf\text{-water}}$ relation, with the highest slope for *Vicia faba* and the lowest slope for *Brassica oleracea* (Fig. 5A) However, we argue that the number of replicates for each plant species is simply too low to interpret this finding robustly. In order to explore possible species-specific effects on the fractionation between the biomarkers and the leaf water in more detail, boxplots of the individual plant species of $\varepsilon_{n\text{-alkane/leaf-water}}$ values are shown in Fig. 5B. Median $\varepsilon_{n\text{-alkane/leaf-water}}$ values are -155‰ for *Brassica oleracea*, -164‰ for *Eucalyptus globulus* and -149‰ for *Vicia faba*, with an overall mean value of -156‰ (ranging from -133 to -192‰). The boxplots of $\varepsilon_{n\text{-alkane/leaf-water}}$ reveal that the median of the three investigated plant species can statistically not be distinguished, due to overlapping notches. Due to the low sample number from each species, the 95% confidence interval is larger than the interquartile range in some cases. However, it seems that at least small species-specific differences cannot be ruled out. Our $\varepsilon_{n\text{-alkane/leaf-water}}$ values resemble well the data from a laboratory study (Kahmen et al., 2011), reporting a median value of -162‰ for $n\text{-}C_{25}$, $n\text{-}C_{27}$ and $n\text{-}C_{29}$ of *Populus trichocarpa*. Furthermore, they are well comparable to climate chamber data of *Betula occidentalis* ($n\text{-}C_{31}$) and *Populus fremontii* ($n\text{-}C_{29}$) from Tipple et al. (2015), reporting a median $\varepsilon_{n\text{-alkane/leaf-water}}$ value of -155‰. In addition, field experiments reveal similar median values of -151‰ (for $n\text{-}C_{29}$) and -142‰ (for $n\text{-}C_{31}$) from typical plant species (excluding grasses) from southern California (Feakins and Sessions, 2010) and -144‰ (for $n\text{-}C_{29}$, of the species *Prunus serotina*, *Acer saccharinum*, *Quercus rubra*, *Quercus alba* and *Ulmus americana*) from the temperate forest at Brown's Lake Bog, Ohio, USA. The large range in $\varepsilon_{xylem\text{-water/leaf-water}}$ values from our study (-192 to -133‰) is also found in different laboratory and field studies (-198 to -115‰, derived from $n\text{-}C_{29}$ and $n\text{-}C_{31}$ data from Feakins and Sessions, 2010; Kahmen et al., 2011a; Tipple et al., 2015; Freimuth et al., 2017).

The observed large range in $\varepsilon_{n\text{-alkane/leaf-water}}$ and the rather low $r^2$ of the relationship between $\delta^2H_{n\text{-alkane}}$ and $\delta^2H_{\text{leaf-water}}$ could point to a more specific water pool being used during biosynthesis rather than bulk leaf water (Sachse et al., 2012; Schmidt et al., 2003). Furthermore, as also NADPH is acting as hydrogen source during $n$-alkane biosynthesis, its $\delta^2H$ is clearly more negative than the biosynthetic water pool (Schmidt et al., 2003), further contributing to a weakening of the $\delta^2H_{n\text{-alkane}}$ to $\delta^2H_{\text{leaf-water}}$ correlation and enlarging the range of $\varepsilon_{n\text{-alkane/leaf-water}}$. In more detail, alkane synthesis takes place by modifying/expanding fatty acids in the cytosol, while fatty acids are synthesized in the chloroplasts (Schmidt et al., 2003). Thus, the cytosol and chloroplast waters are two hydrogen sources. However hydrogen can additionally be added to the alkanes and fatty acids by NADPH which originates from different sources (photosynthesis and pentose phosphate cycle, Schmidt et al., 2003). It is therefore challenging to measure directly the water pool from which the alkanes are biosynthesized (Tipple et al., 2015). Moreover, biosynthetic and metabolic pathways in general (Kahmen et al., 2013; Sessions et al., 1999; Zhang et al., 2009), the carbon and energy metabolism of plants more specifically (Cormier et al., 2018) and the number of carbon atoms of the $n$-alkane chains (Zhou et al., 2010) may have an influence on the fractionation. Our $\varepsilon_{n\text{-alkane/leaf-water}}$ values correlate with $T_{air}$ (Fig. A2A), whereas the correlation with $RH_{air}$ (Fig. A2B) is not significant. This could point to a relationship between $\varepsilon_{xylem\text{-water/leaf-water}}$ and plant physiological processes (affecting various plants differently). In summary, the fractionation between leaf water and $n$-alkanes is strongly influenced by the metabolic pathway of the $n$-alkane biosynthesis including direct hydrogen transfers, exchange reactions and NADPH as hydrogen source.

### 3.2.2 $\delta^{18}O_{sugar}$ vs. $\delta^{18}O_{leaf\text{-water}}$

A highly significant correlation is observed for the correlation between $\delta^{18}O_{sugar}$ and $\delta^{18}O_{leaf\text{-water}}$ ($r^2 = 0.84$, $p < 0.001$; Fig. 6A). The regression reveals a slope of 0.74 and an intercept of 30.7‰. The observed slope of the $\delta^{18}O_{sugar}$ vs. $\delta^{18}O_{leaf\text{-water}}$ relationship could serve as an indicator for a leaf water (heavy-isotope enrichment) signal transfer damping of approximately 26%.

The theory behind the signal damping is adopted from the cellulose research (e.g. Barbour and Farquhar, 2000). Barbour and Farquhar (2000) related the extent of the signal damping to the proportion of unenriched source water, which contributes to the local synthesis water pool and to the proportion of exchangeable oxygen during cellulose synthesis. Our damping of 26% is within the range of values reported for cellulose synthesis in *Gossypium hirsutum* leaves (between 35 and 38%; Barbour and Farquhar, 2000), for *Eucalyptus globulus* leaf samples (38%; Cernusak et al., 2005) and for five $C_3$ and $C_4$ grasses (25%; Helliker and Ehleringer, 2002). Recently, Cheesman and Cernusak (2017) provided damping for leaf cellulose synthesis based on plant data grown under same conditions at Jerusalem Botanical Gardens published by Wang et al. (1998), ranging between 4 and 100% with a mean of 49%, revealing large variations among and between ecological groups (namely conifers, deciduous, evergreen and shrubs). A large range of signal damping associated with leaf cellulose was also reported by Song et al. (2014) for *Ricinus communis* grown under controlled conditions. A common disadvantage of the above-mentioned studies is the absence of direct measurements of the proportion of depleted source water contribution to the local synthesis water (as noticed by Liu et al., 2017), which largely contributes to the extent of the signal damping (Barbour and Farquhar, 2000). However, when transferring cellulose results to pentoses, such as hemicellulose-derived arabinose and xylose, it should be noted that they are biosynthesized via decarboxylation of the carbon at position six (C6) from glucose (Altermatt and Neish, 1956; Burget et al., 2003; Harper and Bar-Peled, 2002). Waterhouse et al. (2013) showed that the oxygen atoms at C6 position in glucose moieties, used for heterotrophic

cellulose synthesis, are strongly affected by the exchange with local water (up to 80%). Based on these findings, it can be suggested that the influence of the non-enriched source water during the synthesis of leaf hemicelluloses is rather small.

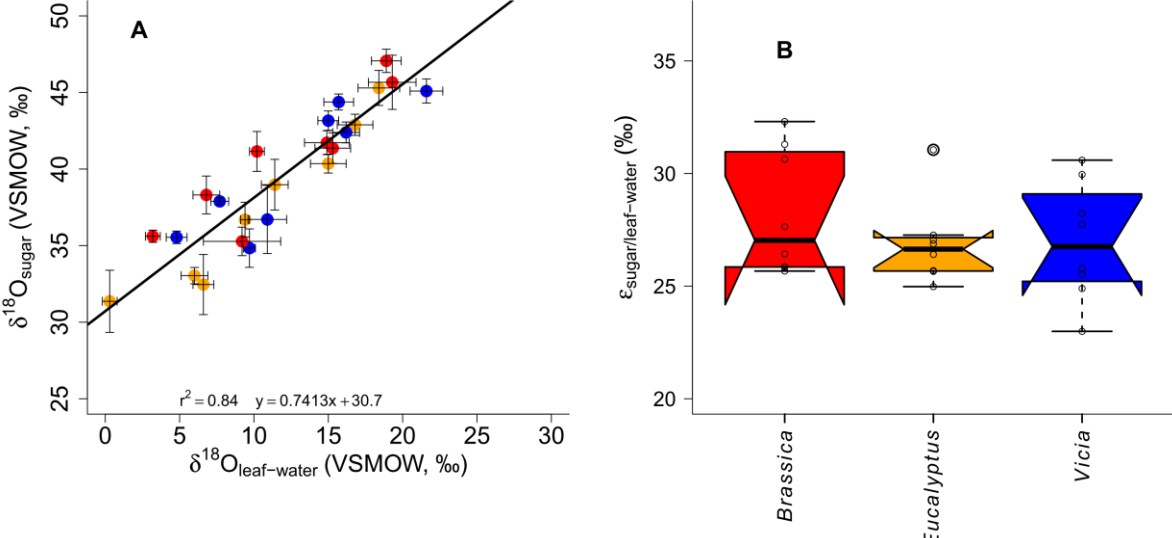

**Fig. 6:** (A) Scatterplot depicting the relationship between $\delta^{18}O_{sugar}$ vs. $\delta^{18}O_{leaf-water}$). Error bars of the $\delta$ values represent standard deviation of repeated measurements (see section 2.2 and Mayr, 2002). (B) Boxplot comprising the plant-specific fractionation $\varepsilon_{sugar/leaf-water}$ according to Eq. 11, showing median (thick black line), interquartile range (IQR) with upper (75%) and lower (25%) quartiles, lower and

450 upper whiskers, which are restricted to $1.5 \cdot$ IQR. Outside the $1.5 \cdot$ IQR space, the data points are marked with a dot. The notches are extend to $\pm 1.58 \cdot IQR/\sqrt{n}$, by convention and give a 95% confidence interval for the difference of two medians (McGill et al., 1978). *Brassica oleracea, Eucalyptus globulus* and *Vicia faba* samples are shown in red, orange and blue, respectively.

The oxygen isotope composition of leaf sugars (e.g. sucrose and cellulose) is strongly influenced by O exchange processes, which cause more positive $\delta^{18}O$ values of sugars compared to leaf water (Barbour et al., 2004; Cernusak et al., 2003). Therefore, median $\varepsilon_{sugar/leaf-water}$ values of +27.0‰ for *Brassica oleracea*, +26.6‰ for *Eucalyptus globulus*, +26.8‰ for *Vicia faba* are observed (Fig. 6B). The overall $\varepsilon_{sugar/leaf-water}$ average value of the three investigated species is +27.3‰ (ranging from +23.0 to +32.3‰).

No systematic difference between the individual species is evident. Moreover, the $\varepsilon_{sugar/leaf-water}$ values do not correlate significantly with $T_{air}$, but significantly with $RH_{air}$ (Fig. A2C and D). A temperature dependence of the $\varepsilon_{sugar/leaf-water}$ is not supported by this experiment, in contrast to results from Sternberg and Ellsworth (2011), where a temperature effect on oxygen fractionation during heterotrophic cellulose biosynthesis is observed. The here observed fractionation between

hemicellulose-derived sugars and leaf water, with regard to $\varepsilon_{sugar/leaf-water}$ values, is well in range with values reported for sucrose (exported from photosynthesizing leaves) and leaf water, which was shown to be +27‰ (Cernusak et al., 2003). The cellulose biosynthesis is also associated with an heavy-isotope enrichment of around +27‰ compared to the synthesis water as shown in growth experiments (Sternberg et al., 1986; Yakir and DeNiro, 1990). The relatively uniform fractionation is explained via

the isotope exchange between the carbonyl oxygens of the organic molecules and the surrounding water (cf. Schmidt et al., 2001). This equilibrium fractionation effect was indeed described earlier by the reversible hydration reaction of acetone in water by Sternberg and DeNiro (1983) to be +28, +28 and +26‰ at 15, 25 and 35°C, respectively. However, the observed range of approximately 9‰ (Fig. 6B) could indicate that partially not only the oxygen equilibrium fractionation between organic

molecules and medium water has to be considered. Presumably, isotope as well as sucrose synthesis gradients within the leaf have to be taken into account when interpreting leaf sugar oxygen isotope compositions and their correlation to leaf water (Lehmann et al., 2017). Lehmann et al. (2017) reported a fractionation between sucrose and leaf water of +33.1‰. Based on this they proposed a conceptual scheme how such gradients can lead to discrepancies between the isotope composition of the bulk leaf water and the synthesis water. Thus, the fractionation between carbohydrates and bulk leaf water can exceed the common average of +27‰. Also Mayr et al. (2015) found a fractionation between aquatic cellulose $\delta^{18}O$ and lake water larger than this value of around +29‰.

### 3.2.3 Inter-species variability

The differences observed in $\varepsilon_{n\text{-alkane/leaf-water}}$ (Fig. 5B) and $\varepsilon_{sugar/leaf-water}$ (Fig. 6B) between the taxa might be explained by different leaf sizes and geometries. Please note that these differences are statistically not significant. We do not know whether this is caused by the rather low sample number of our study (n = 24) or not. Still, the yielded mean biosynthetic fractionation factors over all species are -156‰ and +27‰ for $\delta^2H_{n\text{-alkanes}}$ and $\delta^{18}O_{sugars}$, respectively. This is well in agreement with data from the literature (usually -160‰ and +27‰, respectively).

When reviewing Fig. 2 and 3, we observe differences between chamber 4 and 8 for *Eucalyptus globulus,* which is mainly related to different sampling time of leaves between the two experiments. Because of the diurnal course of light, which was simulated in each chamber, relative humidity and temperature steady-state conditions prevailed only during 11 am and 4 pm (see section 2.1). While the leaf samples of *Eucalyptus globulus* from chambers 5 to 8 were sampled during the simulated steady state daytime conditions, chambers 1 to 4 were sampled after this simulated steady state daytime conditions due to time restrictions, explaining the deviations in leaf water isotope composition. The sampling time, however, does not influence isotope values of tissue samples, which represent the integrated signal over the entire growing period and does not explain the observed differences in $\delta^2H_{n\text{-alkane}}$. The $\delta^2H_{n\text{-alkane}}$ values differ for all plant types whereas $\delta^{18}O_{sugar}$, $\delta^{18}O_{leaf\text{-water}}$ do not. The differences in $\delta^2H_{n\text{-alkane}}$ are most likely explainable via the fractionation occurring during biosynthesis of *n*-alkanes, which is depending not only on leaf water isotope composition but also on plant physiological factors (e.g. water pressure deficit between air and leaf, transpiration rate, assimilation rate; cf. discussion in 3.2.1).

### 3.2.4 Implications for paleoclimatic reconstructions

The damping of the leaf water oxygen isotope signal caused by the exchange between sugars and water is an important issue, especially in stem, trunk and root tissues. While these tissues hardly produce *n*-alkanes, (hemi-)celluloses and sugars extracted from such tissues do not show the full leaf water heavy-isotope-enrichment because of a partial oxygen exchange with non-enriched stem water (e.g. Zech et al., 2014a). This has to be kept in mind for paleoclimatic reconstructions, especially for the sugars of grasses, which do not record the full leaf water enrichment (e.g. Helliker and Ehleringer, 2002b). In the case of grasses, this signal damping affects *n*-alkanes, too. However, such uncertainties can be included in the RH reconstruction via assumptions and sensitivity analysis of the used model, thus allowing quantifying uncertainties of reconstructed RH records as shown by Hepp et al. (2019).

### 3.3 Strong control of relative humidity over deuterium-excess of leaf water

#### 3.3.1 Calculating relative humidity based on measured leaf water

The correlations between leaf water-based and measured $RH_{air}$ and modeled $d_e$ and measured deuterium-excess$_{leaf-water}$ are illustrated in Fig. 7A and B, respectively. Since the climate chambers had a high fresh air supply (see section 2.1) the variability of the leaf water deuterium-excess is predominantly the result of leaf water heavy-isotope enrichment and thus strongly driven by relative humidity.

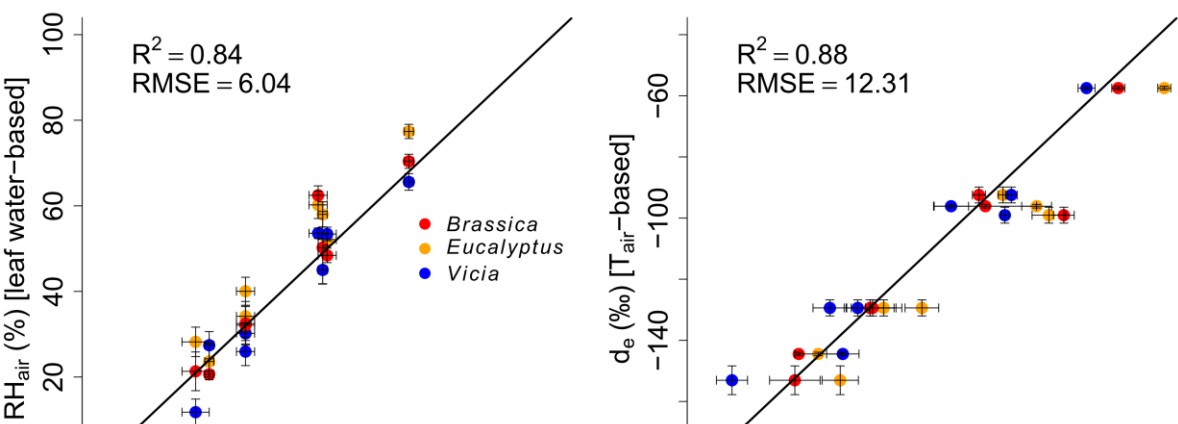

**Fig. 7:** Scatterplots illustrating the correlation between (A) modeled (leaf water-based) and measured air relative humidity ($RH_{air}$) and modeled ($T_{air}$−based) vs. measured leaf water deuterium-excess. Black lines indicate the 1:1 relationship. $R^2$ and RMSE are calculated as described in section 2.4, while the

RMSE values have the dimensions of the respective variables. Error bars for the measured RH values represent analytical standard deviations (see Mayr, 2002). See section 2.4 for the uncertainties of the calculated and modeled results.

Evidence for the strong control of relative humidity on deuterium-excess of leaf water comes from

multivariate regression analysis between the measured deuterium-excess$_{leaf-water}$ values versus $RH_{air}$, and $T_{air}$. The results reveal that the deuterium-excess$_{leaf-water}$ significantly correlates with $RH_{air}$ of the climate chambers (p < 0.001), with an $r^2$ of 0.92. The strong control of RH on the deuterium-excess of leaf water is furthermore supported by the significant correlations between calculated versus measured $RH_{air}$ values (Fig. 7A). This is in line with the strong correlation between modeled $d_e$ based

on $T_{air}$ and measured deuterium-excess$_{leaf-water}$ values (Fig. 7B).

The modeled $d_e$ values show a high agreement with measured deuterium-excess of leaf water without being too positive. This could be expected from the literature, because the output of the Craig-Gordon-based leaf water enrichment model (e.g. Allison et al., 1985; Barbour et al., 2004; Cernusak et al., 2016; section 2.3) reflects leaf water at the evaporative sites, which should be more positive than the

measured bulk leaf water. Especially under low relative humidity conditions, the discrepancy between Craig-Gordon model results and the measured values is shown to be more pronounced. This is associated with higher transpiration fluxes and higher isotope heterogeneity within the leaf water due to a non-uniform closure of the stomata (Flanagan et al., 1991; Santrucek et al., 2007). An overestimation of the Craig-Gordon models can hardly be observed here (Fig. 7B). However, based on

the accepted leaf water enrichment theory (e.g. Cernusak et al., 2016), higher transpiration rates (e.g.

under low humidity conditions) should still lead to a larger discrepancy between Craig-Gordon modelled and measured leaf water, because the back diffusion of enriched leaf water from the evaporative sites should get lower the higher the transpiration flux is.

It should be noted that there is the possibility to build up a more detailed model based on Eq. 2. With this, a model is given to derive $d_e$ values with the usage of $d_a$ and $d_s$, which can be compared to the measured deuterium-excess$_{leaf\text{-}water}$ values. However, when modeling $d_e$ without the simplification $\delta_a - \delta_s \approx -\varepsilon*$ the $R^2$ results to 0.86 and RMSE equals 13.07‰ compared to the presented 0.88 and 12.31‰. Furthermore, in Eq. 5 $T_{air}$ can be replaced by $T_{leaf}$. With this, Eq. 2 results to values based on leaf temperature. This would consider that the Craig-Gordon model requires the temperature of the evaporating surface rather than the air temperature for $e_i$. However, with this model extension the $R^2$ and the RMSE are 0.55 and 23.54‰, respectively. By rearranging Eq. 2, RH values can be modeled which can be compared to $RH_{air}$ as well as $RH_{leaf}$ values ($e_a/e_i$ multiplied by 100 with $T_{leaf}$). The respective model characteristics are again lower for the $RH_{leaf}$ case ($R^2 = 0.27$ and RMSE = 11.84%) than for the $RH_{air}$ comparison ($R^2 = 0.81$ and RMSE = 6.56%). Still Eq. 6 provides better results, as presented in this paragraph ($R^2 = 0.84$ and RMSE = 6.04%). This discussion is in line the differences between $T_{leaf}$ vs. $T_{air}$ and $RH_{leaf}$ vs. $RH_{air}$ conditions in the climate chambers. They reveal the same trends and magnitude, but $T_{leaf}$ is consequently higher than $T_{air}$ along with higher $RH_{leaf}$ values compared to $RH_{air}$ (Fig. 1; Mayr, 2002). Summarized we therefore argue that the model presented in Eq. 6 (including the simplifications of $\delta_a - \delta_s \approx -\varepsilon*$ and using $T_{air}$ in Eq. 5) is able to reconstruct $RH_{air}$ values based on $\delta^2H_{leaf\text{-}water}$ and $\delta^{18}O_{leaf\text{-}water}$ values.

### 3.3.2 Calculating relative humidity based on reconstructed leaf water

In order to test the proposed paleohygrometer approach, the alkane and sugar-based (reconstructed) leaf-water values were used to calculate $RH_{air}$. The measured $RH_{air}$ is well reflected by the biomarker-based air relative air humidity values ($R^2 = 0.54$; Fig. 8). Overall, a lower coefficient of determination of the biomarker-based model compared to the leaf water-based reconstructions (compare black with grey data points in Fig. 8) is observed. This can be attributed to the uncertainties in reconstructed leaf water using $\delta^2H_{n\text{-}alkane}$ and $\delta^{18}O_{sugar}$ datasets as discussed in sections 3.2.1 and 3.2.2. Briefly, the limitations regarding $\delta^2H$ arise from the rather weak relationship between the $\delta^2H$ of the $n$-alkanes and the leaf water, probably linked with the large range in $\varepsilon^2_{n\text{-}alkane/leaf\text{-}water}$ (Fig. 5B). The applied equation for reconstructing $\delta^2H_{leaf\text{-}water}$ by using $\delta^2H_{n\text{-}alkane}$ using a constant biosynthetic fractionation of -160‰ (Eq. 10) was considered to be suitable (Sachse et al., 2012; Sessions et al., 1999). However, this equation also contributes some uncertainty to the final relative humidity reconstruction. With regard to $\delta^{18}O$, the relatively large variations in $\varepsilon_{sugar/leaf\text{-}water}$ of 9‰ have to be considered (Fig. 6B), because in the $\delta^{18}O_{leaf\text{-}water}$ reconstructions a fixed value of +27‰ is used (Eq. 11). Such a uniform biosynthetic fractionation is just an approximation of the real, probably more variable values. When biomarkers are used to derive leaf water, the measured bulk leaf water does not capture the isotope value of the water in which the biomarkers are biosynthesized. This could thus partly explain the weaker relationship between measured and calculated $RH_{air}$ . Especially the underestimation of the biomarker-based $RH_{air}$ values under the 68% relative humidity conditions, as well as the large range in reconstructed $RH_{air}$ values for the 48, 49, 50% $RH_{air}$ chambers can be attributed to the leaf water reconstruction uncertainties. It should be mentioned that using Eqs. 7 and 8 to calculate leaf water isotope composition based on the biomarkers via a biosynthetic fractionation values implies that the fractionation process in principle can be treated as single process with a unique source. While this approximation can be questioned (see discussion in section 3.2), the overall correlation between

biomarker-based and measured RH$_{air}$ highlights the potential of the approach (Hepp et al., 2017; Tuthorn et al., 2015; Zech et al., 2013a), also for future paleoclimate reconstructions.

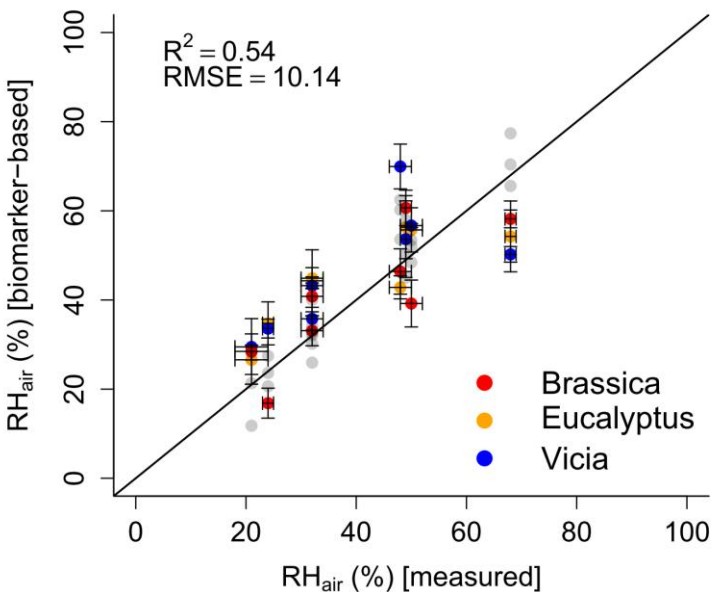

**Fig. 8:** Scatterplot depicting the relationship between modeled (biomarker-based) and measured relative air humidity (RH$_{air}$). The black line indicates the 1:1 relationship. R$^2$ and RMSE was calculated as described in section 2.4, while the RMSE values have the dimensions of the respective variables. Error bars for the measured values represent analytical standard deviations (see Mayr, 2002). For uncertainty calculation of the results, see section 2.4. In addition, the leaf water-based relative air humidity results are shown in light grey for comparison.

### 3.4 Coupling δ$^2$H$_{n\text{-}alkane}$ and δ$^{18}$O$_{sugar}$ for δ$^2$H$_{source\text{-}water}$ and δ$^{18}$O$_{source\text{-}water}$ calculation

The second advantage of the proposed coupled δ$^2$H$_{n\text{-}alkane}$-δ$^{18}$O$_{sugar}$ approach is a more robust reconstruction of the isotope composition of the source water, which can often be directly linked to the local precipitation signal (Hepp et al., 2015, 2017; Tuthorn et al., 2015; Zech et al., 2013a). Therefore, we calculated the source water isotope compositions via the slopes of the LEL's and the GMWL. In order to show the difference of the approaches using either δ$^2$H$_{n\text{-}alkane}$ or δ$^{18}$O$_{sugar}$ for source water reconstruction, figure 9 depicts (i) the leaf water isotope composition reconstructed from δ$^2$H$_{n\text{-}alkane}$ and δ$^{18}$O$_{sugar}$ results using the biosynthetic fractionation factors, (ii) the measured δ$^2$H$_{leaf\text{-}water}$ and δ$^{18}$O$_{leaf\text{-}water}$ values, (iii) the source water isotope composition based on reconstructed leaf water, (iv) the source water isotope composition based on measured leaf water along with the (v) tank water isotope composition.

For δ$^2$H, neither the range nor the median of the δ$^2$H$_{leaf\text{-}water}$ are well captured by the alkane-based leaf water values. However, the overlapping notches do not support a statistical difference in the median values (Fig. 9A). The median is on average 13‰ more positive than the measured δ$^2$H$_{tank\text{-}water}$. A higher agreement between measured and modeled values is observed from leaf water-based δ$^2$H$_s$ compared to δ$^2$H$_{tank\text{-}water}$. The average offset is reduced to 2‰ and the range is reduced by approximately 70‰, compared to the biomarker-based reconstruction.

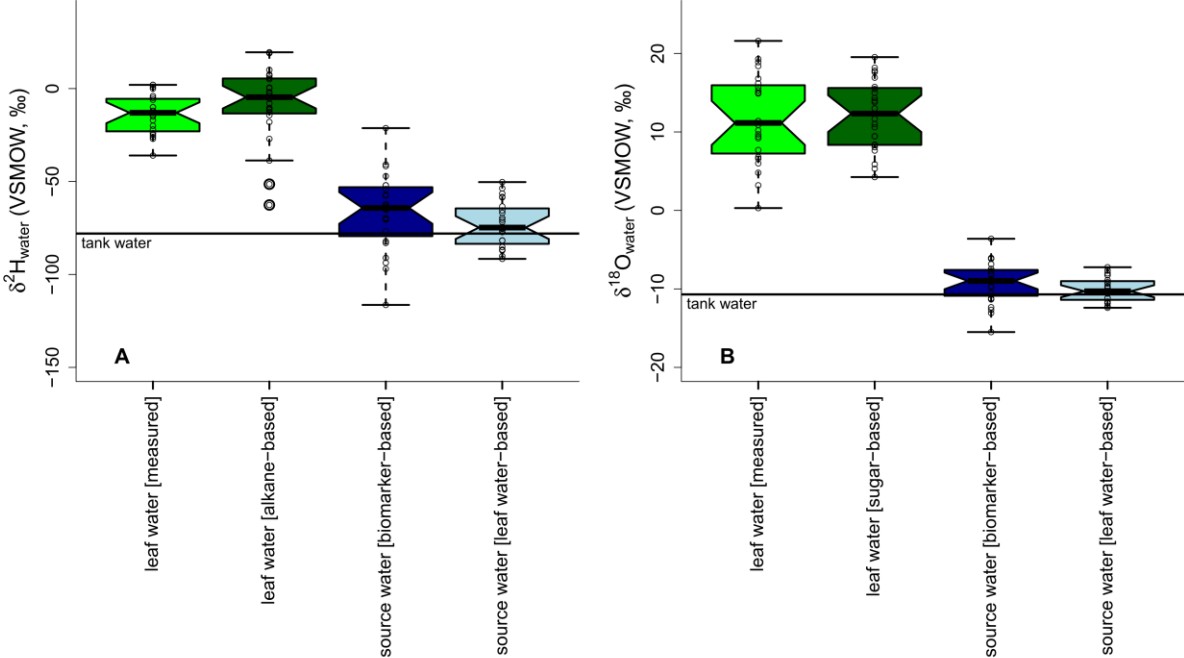

**Fig. 9:** Boxplots showing the measured leaf water in comparison to the biomarker-based leaf water, source water calculated with biomarker-based leaf water values and source water based on measured leaf water (A: $\delta^2H_{leaf-water}$; B: $\delta^{18}O_{leaf-water}$). Tank water is given as a black solid line in both plots. Source water isotope compositions were calculated via the slopes of the LEL′s (either with biomarker-based or measured leaf water values) and the GMWL. Boxplots show median (thick black line), interquartile range (IQR) with upper (75%) and lower (25%) quartiles, lower and upper whiskers, which are restricted to $1.5 \cdot$ IQR. Outside the $1.5 \cdot$ IQR space, the data points are marked with a dot. The notches are extend to $\pm 1.58 \cdot IQR/\sqrt{n}$, by convention and give a 95% confidence interval for the difference of two medians (McGill et al., 1978).

For $\delta^{18}O$, the sugar-based leaf water values agree with the measured ones regarding the median values, as supported by the largely overlapping notches (Fig. 9B). The range of the reconstructed leaf water is about 6‰ lower than for the measured $\delta^{18}O_{leaf-water}$ dataset. All reconstructed $\delta^{18}O_s$ values, regardless whether they are biomarker- or leaf water-based, are comparable to the measured $\delta^{18}O_{tank-water}$. While the biomarker-based datasets depict an average offset of 2‰, the leaf water-based values only differ by 0.3‰ from the tank water $\delta^{18}O$ values, referring to the median.

The overall larger range of modeled $\delta^2H_s$ and $\delta^{18}O_s$ compared to measured $\delta^2H_{tank-water}$ and $\delta^{18}O_{tank-water}$ can be related to uncertainties in $S_{LEL}$ modeling (see equations in section 2.3.2). Bariac et al. (1994) found no agreement between the intersects of modeled LEL′s with the GMWL and the plant source water. Allison et al. (1985) explained such results with changing environmental conditions, leading to various LEL′s with a locus line not necessarily passing the $\delta^2H_s$ and $\delta^{18}O_s$ data point, in a system that approaches rapidly new steady-state conditions. However, when a larger isotope range of source water is used in the studies, leaf water seems to be a good tracer for source water changes when a Craig-Gordon model is applied (Benettin et al., 2021; Bush et al., 2017).

## 4 Conclusions

The climate chamber results suggest that leaf wax-derived $n$-alkane and hemicellulose-derived sugar biomarkers are valuable $\delta^2H_{\text{leaf-water}}$ and $\delta^{18}O_{\text{leaf-water}}$ recorders, respectively. The coupling of $\delta^2H_{n\text{-alkane}}$ and $\delta^{18}O_{\text{sugar}}$ by using a simplified Craig-Gordon equation allows moreover a more robust $RH_{\text{air}}$ reconstruction of the chambers in which the plants were grown compared to single isotopes. With regard to the research questions, we conclude the following:

(i)   $n$-C$_{29}$ predominated and occurred at abundances suitable for compound-specific $\delta^2H$ measurements in the leaf samples from all investigated species (*Eucalyptus globulus*, *Vicia faba* var. *minor* and *Brassica oleracea* var. *medullosa*). For *Vicia faba*, additionally $n$-C$_{31}$ could be evaluated robustly. $\delta^{18}O_{\text{sugar}}$ values could be obtained for the hemicellulose-derived monosaccharides arabinose and xylose.

(ii)  Both the $\delta^2H_{n\text{-alkane}}$ and $\delta^{18}O_{\text{sugar}}$ values yielded highly significant correlations with $\delta^2H_{\text{leaf-water}}$ and $\delta^{18}O_{\text{leaf-water}}$ ($r^2$ = 0.45 and 0.85, respectively; $p < 0.001$, $n$ = 24). Mean fractionation factors between biomarkers and leaf water were found to be -156‰ (ranging from -133 to -192‰) for $\varepsilon_{n\text{-alkane/leaf-water}}$ and +27.3‰ (ranging from +23.0 to +32.3‰) for $\varepsilon_{\text{sugar/leaf-water}}$.

(iii) $RH_{\text{air}}$ can be derived robustly by using the measured leaf water isotope composition ($\delta^2H_{\text{leaf-water}}$ and $\delta^{18}O_{\text{leaf-water}}$) and a rearranged Craig-Gordon model, ($R^2$ = 0.84; $p < 0.001$; RMSE = 6%).

(iv)  Biomarker-based and measured $RH_{\text{air}}$ correlation with $R^2$ of 0.54 ($p < 0.001$) and RMSE of 10% highlights the great potential of the coupled $\delta^2H_{n\text{-alkane}}$-$\delta^{18}O_{\text{sugar}}$ paleohygrometer approach for reliable relative humidity reconstructions. Uncertainties regarding relative humidity reconstructions via biomarker-based leaf water isotope composition arose from leaf water reconstructions and model uncertainties, as shown in the conclusions ii) and iii).

(v)   The coupled $\delta^2H$-$\delta^{18}O$ approach enables a better back calculation of the plant source water compared to single isotopes. Reconstructed source water isotope composition ($\delta^2H_s$, $\delta^{18}O_s$) is in range with the measured tank water ($\delta^2H_{\text{tank-water}}$, $\delta^{18}O_{\text{tank-water}}$). However, modeled $\delta^2H_s$ and $\delta^{18}O_s$ showed a clearly larger range compared to $\delta^2H_{\text{tank-water}}$ and $\delta^{18}O_{\text{tank-water}}$. The uncertainties for source water determination are thus considerably higher compared to the relative humidity reconstructions.

## Acknowledgements

We would like to thank M. Bliedtner and J. Zech (both University of Bern) for help during lipid biomarker and $\delta^2H_{n\text{-alkane}}$ analysis. We thank M. Benesch (Martin-Luther-University Halle-Wittenberg) and M. Schaarschmidt (University of Bayreuth) for laboratory assistance during sugar biomarker and $\delta^{18}O_{\text{sugar}}$ analysis. The research was partly funded by the Swiss National Science Foundation (PP00P2 150590). We also acknowledge N. Orlowski (University of Freiburg), M. M. Lehmann (Swiss Federal Institute WSL, Birmensdorf) and L. Wüthrich (University of Bern) for helpful discussions. We are very grateful for the constructive discussion on an earlier version of this manuscript as preprint at Biogeosciences Discussions (https://doi.org/10.5194/bg-2019-427). We cordially thank Helge Niemann and two anonymous reviewers for the great editorial support and their constructive and encouraging reviews. Involvement of K. Rozanski was supported by Polish Ministry of Science and Higher Education, project no. 16.16.220.842 B02. J. Hepp greatly acknowledges the support given by the German Federal Environmental Foundation. The experiment carried out by C. Mayr was gratefully supported by the HGF-project "Natural climate variations from 10,000 years to the present" (project no. 01SF9813). The experiments were possible due to the assistance of J.B. Winkler, H. Lowag, D.

Strube, A. Kruse, D. Arthofer, H. Seidlitz, D. Schneider, H. D. Payer, and other members of the Helmholtz Zentrum München. This publication was funded by the German Research Foundation (DFG) and the University of Bayreuth in the funding programme Open Access Publishing.

## Author contributions

J. Hepp, M. Zech and C. Mayr wrote the paper; C. Mayr was responsible for the climate chamber experiment together with W. Stichler and provided the leaf samples and the data; M. Zech and R. Zech were responsible for compound-specific isotope analysis on the biomarkers; J. Hepp, M. Tuthorn and I. K. Schäfer did laboratory work and data evaluation of the biomarker compound-specific isotope analysis; B. Glaser, D. Juchelka, K. Rozanski and all co-authors contributed to the discussion and commented on the manuscript.

## Appendix

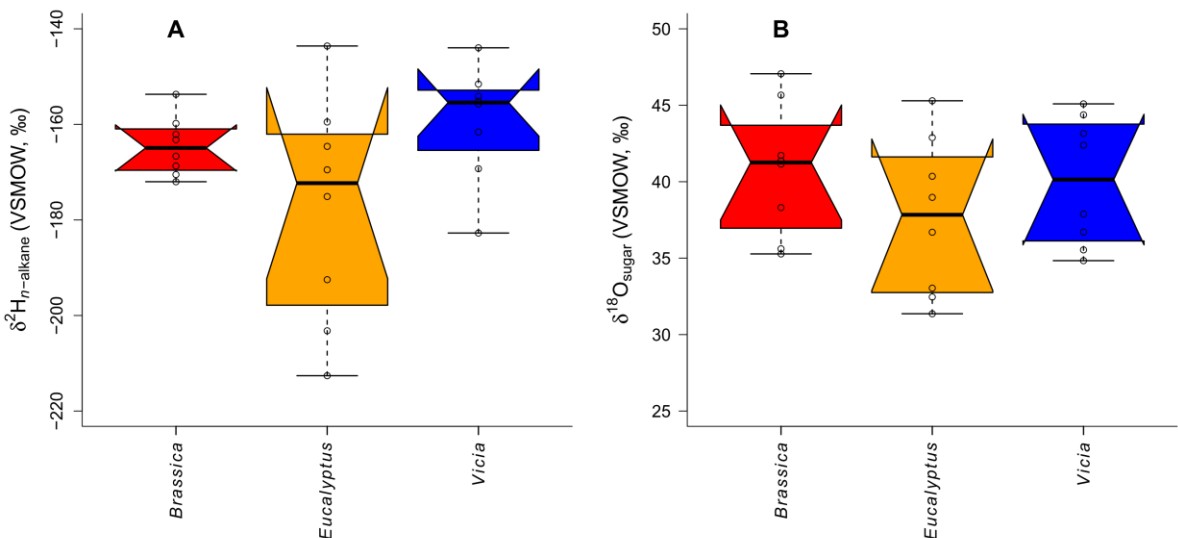

**Fig. A1:** Boxplots comprising the plant-specific $\delta^2H_{n\text{-alkane}}$ (A) and $\delta^{18}O_{sugar}$ values (B). *Brassica oleracera, Eucalyptus globulus* and *Vicia faba* samples are shown in red, orange and blue, respectively. Boxplots show median (thick black line), interquartile range (IQR) with upper (75%) and lower (25%) quartiles, lower and upper whiskers, which are restricted to $1.5 \cdot IQR$. Outside the $1.5 \cdot IQR$ space, the data points are marked with a dot. The notches are extend to $\pm 1.58 \cdot IQR/\sqrt{n}$, by convention and give a 95% confidence interval for the difference of two medians (McGill et al., 1978).

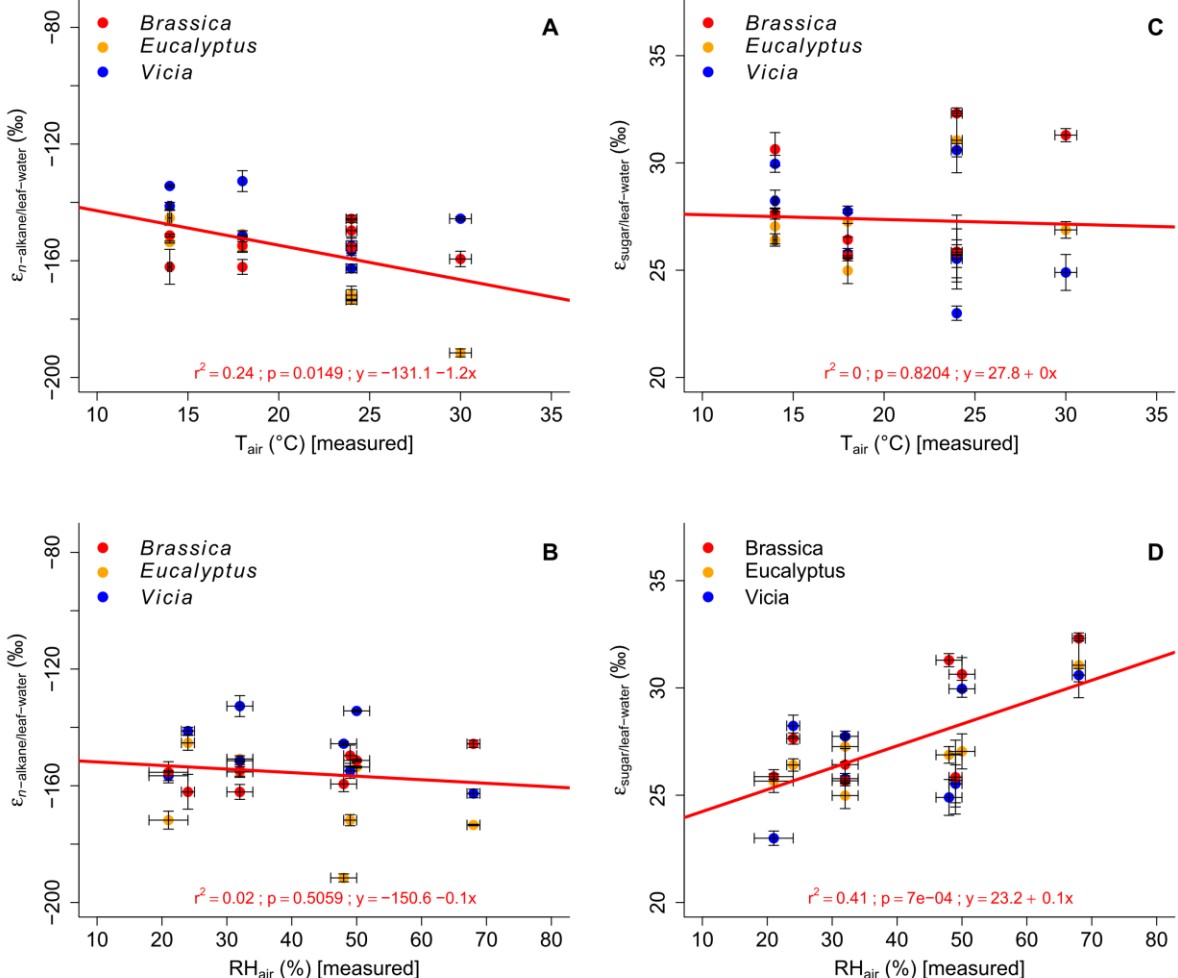

**Fig. A2:** Scatterplots of the fractionation between the biomarkers and leaf water vs. air temperature, air relative humidity (A and B: $\varepsilon_{n\text{-alkane/leaf-water}}$ according Eq. 10; C and D $\varepsilon_{\text{sugar/leaf-water}}$ according Eq. 11). *Brassica oleracera, Eucalyptus globulus* and *Vicia faba* samples are shown in red, orange and blue, respectively. Error bars for the measured values represent analytical standard deviations of repeated measurements (see section 2.2 and Mayr, 2002). For uncertainty calculation of the $\varepsilon$ values, see section 2.4.

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
