# Peer review of "Validation of a coupled $\delta^2 H_{n\text{-alkane}} - \delta^{18} O_{\text{sugar}}$ paleohygrometer approach based on a climate chamber experiment"

_Biogeosciences, 2020_

## Referee Comment (RC1) · Anonymous Referee #1 · 18 Jan 2021

Review of BG-2020-434 Validation of a coupled $\delta$2Hn-alkane-$\delta$18Osugar paleohygrometer approach based on a climate chamber experiment Johannes Hepp, Christoph Mayr, Kazimierz Rozanski, Imke Kathrin Schäfer, Mario Tuthorn, Bruno Glaser, Dieter Juchelka, Willibald Stichler, Roland Zech and Michael Zech.

Dear associate editor, Johannes Hepp and co-authors,

To start with my conclusion, I think this should be published, it is a beautiful dataset and I have the feeling that this dual isotope method for paleo reconstructions will become more and more important. I do have some issues with the manuscript as is. I had difficulties keeping track of the story a little, it is complex material, but I have the idea

the authors have added to, at least, my confusion. I will try to explain what I mean. The measured data clearly shows the isotopic link (similarity) between source water, soil water and xylem water and leaf water moving away from the source water. This leaf water is assumed to be the source for biosynthesis, to some extend at least and on top of that there is biological isotope fractionation. The idea of using both isotopes is that hydrogen isotopes are highly affected by biological fractionation and oxygen isotopes not or less. One reason for that, I read would be O exchange between hemicellulose and leaf water. I had hoped oxygen would also be less dependent on leaf water evap-oration, which it could be in paleorecords were bulk soils will be used and not only leaves. Which made me wonder how leaf derived sugars compare to stem or whole plant derived sugars? Especially for the paleo applications the authors mention as a reason for simplifying some of the formulas.

I have noticed that there is measured leaf water, but also sometimes leaf water isotope values reconstructed based on measured n-alkanes, especially in the comparisons with literature data. On top of that there is evaporative site water isotopic composition especially for H isotopes. At least that was my impression. There is also biosynthetic water that might be different, or water at the site of biosynthesis, again especially for hydrogen. Cytosol versus chloroplast. The role of NADPH is also mentioned. This helps explain the variability, but a lot falls under biosynthetic fractionation, so it explains variability in fractionation. It makes sense all these different versions of water and their impact, but the way the manuscript is organized no I find it extremely confusing. I would suggest a discussion along the main lines ending with some possible explanations for the "scatter". One of the reasons I ask this is because apparently this is also going on for oxygen, leaf water isn't leaf water for oxygens isotopes either. There are isotopic and sucrose synthesis gradients that have o be taken into account. Again, it makes sense, but it is very confusing. And what does this mean for paleo applications? Where are the sugars coming from in paleosols? How is that related to this study? In general, what would be the effect of the oxygen exchange mentioned in the manuscript on paleo samples?

The authors mention that incubation 4 and 8 are the same, or the climate rooms are, the results are not? Any ideas what might be the reason? Some of the biosynthetic or synthesis water issues mentioned above? Or something different? Already the measured leaf water isotopes are different especially for Eucalyptus.

The authors calculate deuterium excess, I have been told that in highly evaporative systems also the slope of the meteoric waterline could be lower than 8. Could that be applicable to these kinds of systems as well?

Specific comments:

It would be great if the symbols in figure 1 could be a bit larger, the difference between squares and triangles is almost in visible.

Would it be possible to indicate the T and RH of the different chambers at least in the figure legend, but if possible, in the figure itself?

In figure 2 it would ne nice if d was defined in the legend.

The per mil in the introduction is fine, the other two I would replace with ‰

Line 226: n-alkane and sugar biomarkers Line 246: from the latter Line 247: ones or values, not once. Line 278: all three plant spp. Line 415: This could point. . .. I have a hard time connecting "this" to the previous sentences. I got lost here a little. Line 472: despite without ? Line 472/473: bulk leaf is less enriched than the leaf water at the evaporative sites. Confusing. Bulk leaf water? The measured leaf water is bulk leaf water I assume? Why not use "leaf water" for this and call the other water, "water at the evaporative sites within the leaf" or something similar. Line 506: This is a very weird way to refer to a figure. You made the figure, it is not measured or observed and therefore a piece of evidence in your reasoning. The sentence starting at therefore till LEL's) can be deleted, I think. Line 550: Or the fact that bulk leaf water as measured does not capture the variability of water within the leaf and potentially important for biosynthesis. At least that has been mentioned several times already.

I do think fractionation and variability therein is important, but the authors discussed these different leaf water to extensive to not mention here. Line 553: introduces

---

## Referee Comment (RC2) · Anonymous Referee #2 · 22 Jan 2021

Hepp et al. describe a project that focuses on investigating the role of environmental conditions on stable isotopic composition of modern plant biomarkers. The authors grew three higher plant species in growth chambers under controlled conditions (varied temperature and relative humidity) and then measured the d2H values of leaf-wax n-alkanes and the d18O values of hemicellulose sugars. These data were supplemented by isotope data (oxygen and hydrogen) for soil, xylem and leaf waters. The goal of the project was to investigate the usefulness of integrating the n-alkane d2H and sugar d18O data for reconstructing paleo-humidity. This is a very detailed and well-described study that is likely to attract attention from biogeochemists, paleoecologists and paleoclimatologists who use biomarker isotopes for studying climate change in the past.

The manuscript fits the scope of Biogeosciences quite well, however, there are several issues I would like to be resolved before this work is published. For this manuscript to be attractive to the readers of this journal, the authors need to have a more detailed discussion of what the implications of their findings are and, more importantly, how the information generated in this project could be used in the paleo context.

First: the applicability of this approach to paleo records The authors do a very thorough job providing a theoretical rationale for their approach and demonstrating that the biomarker data they've obtained can be used to reconstruct the isotopic composition of leaf water. Moreover, the dual biomarker approach could work well for estimating paleo relative humidity. However, it is not clear how exactly this approach would work with sedimentary n-alkanes and sugars, provided the latter survive the fossilisation process. What type of sedimentary material would be needed with this approach, i.e. would it be intact plant fossils or would this work on dispersed particular plant matter as well?

Second: the choice of plants The authors have used three very different – morphologically and physiologically speaking – higher plants without really explaining why they chose these particular plants. What guided their choice and why do they think the isotope data they've generated (and the fractionation factors they've calculated for these species) are representative of how the dual $d2H – d18O$ approach works in general in higher plants. The species-specific hydrogen isotope fractionation between leaf water and biomarkers shown in Fig. 4 indicates that there are considerable differences among the three species. At the same time, the authors state on lines 99-100 "In case the biosynthetic fractionation is known and constant, there is a great potential to derive $RH_{air}$ from coupling $d2H_{n-alkane}$ with $d18O_{sugar}$ values." But for paleo samples, we wouldn't really know what the fractionation was. How would we deal with this issue when looking at a paleo record?

Several other suggestions regarding the quality of figures:

Figure 1. This figure is very busy. It needs to be split into 3 figures (climate chamber

conditions, d2H of waters and n-alkanes, and d18O of waters and sugars). At this scale, it is difficult to see what is going on.

Figure 2. The font in the legend is too small. I expect that I won't be legible when transferred to a published figure. Also, when showing biomarkers, it would be useful to identify the three different plants used for this study, like is done in Figure 3.

Figure 3. The choice of symbol colours when showing different plants could be improved. At this scale, it is difficult to see the difference between purple and black. The symbols themselves could also be made larger.

Figure 6. The same issue as mentioned for Figure 1, i.e., the figure should be split into two figures to make the text more legible. Also, does it make sense to show tank water data as a "box"? It would look better if plotted as a single data point.

---

## Author Comment (AC1) · 23 Feb 2021

**Reply to Referee #1**

by Johannes Hepp, Michael Zech, Christoph Mayr & co-authors

*Dear associate editor, Johannes Hepp and co-authors,*

*To start with my conclusion, I think this should be published, it is a beautiful dataset and I have the feeling that this dual isotope method for paleo reconstructions will become more and more important. I do have some issues with the manuscript as is. I had difficulties keeping track of the story a little, it is complex material, but I have the idea the authors have added to, at least, my confusion. I will try to explain what I mean. The measured data clearly shows the isotopic link (similarity) between source water, soil water and xylem water and leaf water moving away from the source water. This leaf water is assumed to be the source for biosynthesis, to some extend at least and on top of that there is biological isotope fractionation. The idea of using both isotopes is that hydrogen isotopes are highly affected by biological fractionation and oxygen isotopes not or less. One reason for that, I read would be O exchange between hemicellulose and leaf water. I had hoped oxygen would also be less dependent on leaf water evaporation, which it could be in paleorecords were bulk soils will be used and not only leaves. Which made me wonder how leaf derived sugars compare to stem or whole plant derived sugars? Especially for the paleo applications the authors mention as a reason for simplifying some of the formulas.*

→ We are very grateful to anonymous Referee #1 for her/his encouraging words concerning the potential and value our coupled $\delta^2 H_{n\text{-alkane}}$-$\delta^{18}O_{sugar}$ paleohygrometer approach and the here presented dataset. The idea of using both isotopes is the possibility to reconstruct leaf water that is isotopically both $^2$H- and $^{18}$O-enriched due to primarily RH-dependent transpiration. This leaf water (i) plots along an evaporation line right of the global meteoric water line in a $^2$H-$^{18}$O diagram and (ii) its isotope signal is incorporated in newly biosynthesized molecules with an in approximation constant biosynthetic fractionation factor. The fractionation between leaf water and *n*-alkanes is strongly influenced by the metabolic pathway of the *n*-alkane biosynthesis including direct hydrogen transfers, exchange reactions, NADPH as hydrogen source (Schmidt et al., 2003, also for more details). Similarly, the oxygen isotopic composition of leaf sugars (sucrose, hemicellulose and cellulose) is strongly influenced by O exchange processes, which cause more positive $\delta^{18}O$ values of sugars compared to leaf water.

We fully agree with Reviewer#1 that O-exchange and thus 'signal damping' is an important issue especially in stem, trunk and root tissues. While these tissues hardly produce *n*-alkanes, (hemi-)celluloses and sugars extracted from such tissues do not show the full leaf water enrichment signal because a partial oxygen exchange with non-enriched stem water (e.g. Zech et al., 2014). We also fully agree with Reviewer'1 that this has to be kept in mind for paleo applications Importantly, this holds also true for the sugars of grasses, which do not record the full leaf water enrichment (e.g. Helliker and Ehleringer, 2002). In the case of grasses, this 'signal damping' affects *n*-alkanes, too. However, such uncertainties can be included in the RH reconstruction via assumptions and sensitivity analysis of the used model, thus allowing quantifying uncertainties of reconstructed RH records as shown e.g. in Hepp et al. (2019).

We will readily and thoroughly check our manuscript during revision in order to avoid possible confusions and to be as clear with our statements as possible.

*I have noticed that there is measured leaf water, but also sometimes leaf water isotope values reconstructed based on measured n-alkanes, especially in the comparisons with literature data. On top of that there is evaporative site water isotopic composition especially for H isotopes. At least that was my impression. There is also biosynthetic water that might be different, or water at the site of biosynthesis, again especially for hydrogen. Cytosol versus chloroplast. The role of NADPH is also mentioned. This helps explain the variability, but a lot falls under biosynthetic fractionation, so it explains variability in fractionation. It makes sense all these different versions of water and their impact, but the way the manuscript is organized now I find it extremely confusing. I would suggest a discussion along the main lines ending with some possible explanations for the "scatter". One of the reasons I ask this is because apparently this is also going on for oxygen, leaf water isn't leaf water for oxygens isotopes either. There are isotopic and sucrose synthesis gradients that have to be taken into account. Again, it makes sense, but it is very confusing. And what does this mean for paleo applications? Where are the sugars coming from in paleosols? How is that related to this study? In general, what would be the effect of the oxygen exchange mentioned in the manuscript on paleosamples?*

→ For hydrogen, as mentioned in the reply above, we will add more clarity to the discussion especially about the fractionation process during the biosynthesis, which cause the scatter.

We will readily reorganize the whole discussion chapter as suggested in order to make it easier to follow. For oxygen, we will also carefully check and simplify our manuscript where possible in order to avoid confusions.

Readily we will address more explicitly the relevant issues for paleo applications during revision as requested by Reviewer#1.

*The authors mention that incubation 4 and 8 are the same, or the climate rooms are, the results are not? Any ideas what might be the reason? Some of the biosynthetic or synthesis water issues mentioned above? Or something different? Already the measured leaf water isotopes are different especially for Eucalyptus.*

→ Thank you for raising this issue. Indeed, the climate conditions for experiment 4 and 8 were the same. The experimental run was run twice because only four chambers were available at the same time and replication was needed. Reviewing the data of figure 1 we observe differences between experiment 4 and 8 for *Eucalyptus globulus* which is mainly related to different sampling daytime of leaves between the two experiments. As a diurnal course of light, relative humidity and temperature was simulated in each chamber, steady-state conditions prevailed only during 11 am and 4 pm. In experiments 1-4 the leaf samples were sampled due to time restrictions after that daytime explaining the deviations between experiment 4 and 8 in leaf water isotope composition only. We will explain this in detail in the revised version. This, however, did not influence isotope values of tissue samples which represent the integrated

signal over the entire growing period and does not explain the observed differences in $\delta^2H_{n\text{-}alkane}$. It became also obvious that the $\delta^2H_{n\text{-}alkane}$ values show this difference for all plant types whereas $\delta^{18}O_{sugar}$, $\delta^{18}O_{leaf\text{-}water}$ do not show this as well as $\delta^2H_{leaf\text{-}water}$ for *Vicia faba* and *Brassica oleracea*. The differences in $\delta^2H_{n\text{-}alkane}$ are most likely explainable via the fractionation occurring during biosynthesis of *n*-alkanes, which is (as stated in the first reply) depending not only on leaf water also on plant physiological factors (e.g. water pressure deficit between air and leaf, transpiration rate, assimilation rate, from Schmidt et al., 2003).

We will add this explanation in the reorganized discussion chapter when discussing the scatter in $\delta^2H_{n\text{-}alkane}$ vs. $\delta^2H_{leaf\text{-}water}$.

*The authors calculate deuterium excess, I have been told that in highly evaporative systems also the slope of the meteoric waterline could be lower than 8. Could that be applicable to these kinds of systems as well?*

→ Yes, we fully agree. Meteoric water lines different from the GMWL (and thus variability of d-excess of precipitation) will certainly affect reconstructions based on our coupled $\delta^2H_{n\text{-}alkane}$-$\delta^{18}O_{sugar}$ paleohygrometer approach. In the experiment, however, such ambient conditions did not prevail. Both the tank water used for irrigation and the air humidity were regularly sampled and plot on the GMWL. The chambers had a high fresh air supply rate (750 m³ h⁻¹), so an evaporative isotope enrichment in the chamber was avoided. Only a slight heavy isotope enrichment in the soil water was observed. Thus, the excess variability primarily is the result of leaf water isotopic enrichment.

*Specific comments:*

*It would be great if the symbols in figure 1 could be a bit larger, the difference between squares and triangles is almost in visible. Would it be possible to indicate the T and RH of the different chambers at least in the figure legend, but if possible, in the figure itself?*

→ Will readily be changed.

*In figure 2 it would be nice if d was defined in the legend. The per mil in the introduction is fine, the other two I would replace with ‰*

→ Will readily be changed.

*Line 226: n-alkane and sugar biomarkers*

→ Will readily be changed.

*Line 246: from the latter*

→ Will readily be changed.

*Line 247: ones or values, not once.*

→ Will readily be changed.

*Line 278: all three plant spp.*

→ Will readily be changed.

*Line 415: This could point.... I have a hard time connecting "this" to the previous sentences. I got lost here a little.*

→ Will readily be changed during the reorganization of the discussion.

*Line472: despite without?*

→ We will delete "despite".

*Line 472/473: bulk leaf is less enriched than the leaf water at the evaporative sites. Confusing. Bulk leaf water? The measured leaf water is bulk leaf water I assume? Why not use "leaf water" for this and call the other water, "water at the evaporative sites within the leaf" or something similar.*

→ Yes, the measured leaf water is bulk leaf water. Will readily be changed during revision in the whole manuscript.

*Line 506: This is a very weird way to refer to a figure. You made the figure, it is not measured or observed and therefore a piece of evidence in your reasoning. The sentence starting at therefore till LEL's) can be deleted, I think.*

→ Will readily be changed.

*Line 550: Or the fact that bulk leaf water as measured does not capture the variability of water within the leaf and potentially important for biosynthesis. At least that has been mentioned several times already. I do think fractionation and variability therein is important, but the authors discussed these different leaf water to extensive to not mention here.*

→ Will readily be changed.

*Line 553: introduces*

→ Will readily be changed.

Literature

Helliker, B. R. and Ehleringer, J. R.: Grass blades as tree rings: environmentally induced changes in the oxygen isotope ratio of cellulose along the length of grass blades, New Phytologist, 155, 417–424, 2002.

Hepp, J., Wüthrich, L., Bromm, T., Bliedtner, M., Schäfer, I. K., Glaser, B., Rozanski, K., Sirocko, F., Zech, R. and Zech, M.: How dry was the Younger Dryas? Evidence from a coupled $\delta^2H$–$\delta^{18}O$ biomarker paleohygrometer applied to the Gemündener Maar sediments, Western Eifel, Germany, Climate of the Past, 15, 713–733, doi:10.5194/cp-15-713-2019, 2019.

Schmidt, H.-L., Werner, R. A. and Eisenreich, W.: Systematics of $^2H$ patterns in natural

compounds and its importance for the elucidation of biosynthetic pathways, Phytochemistry Reviews, 2(1–2), 61–85, doi:10.1023/B:PHYT.0000004185.92648.ae, 2003.

Zech, M., Mayr, C., Tuthorn, M., Leiber-Sauheitl, K. and Glaser, B.: Oxygen isotope ratios ($^{18}O/^{16}O$) of hemicellulose-derived sugar biomarkers in plants, soils and sediments as paleoclimate proxy I: Insight from a climate chamber experiment, Geochimica et Cosmochimica Acta, 126(0), 614–623, doi:http://dx.doi.org/10.1016/j.gca.2013.10.048, 2014.

---

## Author Comment (AC2) · 23 Feb 2021

**Reply to Referee #2**

by Johannes Hepp, Michael Zech & co-authors

*Hepp et al. describe a project that focuses on investigating the role of environmental conditions on stable isotopic composition of modern plant biomarkers. The authors grew three higher plant species in growth chambers under controlled conditions (varied temperature and relative humidity) and then measured the d2H values of leaf-wax n-alkanes and the d18O values of hemicellulose sugars. These data were supplemented by isotope data (oxygen and hydrogen) for soil, xylem and leaf waters. The goal of the project was to investigate the usefulness of integrating the n-alkane d2H and sugard18O data for reconstructing paleo-humidity. This is a very detailed and well-describedstudy that is likely to attract attention from biogeochemists, paleoecologists and pale-oclimatologists who use biomarker isotopes for studying climate change in the past.*

→ We are very grateful to anonymous Referee #2 for her/his encouraging words concerning the attraction of our study and manuscript for different biogeoscientifically working communities.

*The manuscript fits the scope of Biogeosciences quite well, however, there are several issues I would like to be resolved before this work is published. For this manuscript to be attractive to the readers of this journal, the authors need to have a more detailed discussion of what the implications of their findings are and, more importantly, how the information generated in this project could be used in the paleo context.*

*First: the applicability of this approach to paleo records. The authors do a very thorough job providing a theoretical rationale for their approach and demonstrating that the biomarker data they've obtained can be used to reconstruct the isotopic composition of leaf water. Moreover, the dual biomarker approach could work well for estimating paleo relative humidity. However, it is not clear how exactly this approach would work with sedimentary n-alkanes and sugars, provided the latter survive the fossilisation process. What type of sedimentary material would be needed with this approach, i.e. would it be intact plant fossils or would this work on dispersed particular plant matter as well?*

→ Thank you for raising that issue about the paleo application of the here validated coupled $\delta^2H_{n-alkane}$-$\delta^{18}O_{sugar}$ paleohygrometer approach. Indeed, there are already some first paleo applications of this approach (Hepp et al., 2017, 2019; Zech et al., 2013) as well as first climate transect validation studies (Hepp et al., 2020; Tuthorn et al., 2015) published. Similarly, the stability of the biomarker isotope signals during degradation was studied amongst others by our working group (Zech et al., 2011, 2012).

In brief, *n*-alkanes and sugars can be extracted compound specifically investigated from plants, soils and a wide range of different sediments; so no intact plant tissues of fossilized leaves are necessary. When using lake sediments a thorough terrestrial versus aquatic source identification of the biomarkers should be made in order to know whether leaf water or lake water are reconstructed. We will gladly include during the revision the existing paleo applications and the paleo applicability in general.

*Second: the choice of plants The authors have used three very different – morphologically and physiologically speaking – higher plants without really explaining why they chose these particular plants. What guided their choice and why do they think the isotope data they've generated (and the fractionation factors they've calculated for these species) are representative of how the dual d2H – d18O approach works in general in higher plants. The species-specific hydrogen isotope fractionation between leaf water and biomarkers shown in Fig. 4 indicates that there are considerable differences among the three species. At the same time, the authors state on lines 99-100 "In case the biosynthetic fractionation is known and constant, there is a great potential to derive RHair from coupling d2Hn-alkane with d18Osugar values." But for paleo samples, we wouldn't really know what the fractionation was. How would we deal with this issue when looking at a paleo record?*

→ The three species were primarily chosen because they are very different. This allows best to check for species-dependencies. An additional criterion was the resilience of the taxa to the climatic conditions and sufficient growth during the experiments. Differences between the taxa might be explained by different leaf sizes and geometries, possibly affecting, e.g., the Peclet effect. But please note that the differences in biosynthetic fractionation shown in Fig. 4 are statistically not significant. We do not know whether this is caused by the rather low sample number of our study (n = 24) or not. Still, the yielded mean biosynthetic fractionation factors over all species are -156‰ and +27‰ for $\delta^2H$ of *n*-alkanes and $\delta^{18}O$ of sugars, respectively. This is well in agreement with data from the literature (usually -160‰ and +27‰, respectively). Hence, for paleo applications it seems well justified to assume in approximation constant biosynthetic fractionation factors (at least for oxygen) when reconstructing leaf water isotopic composition from *n*-alkane and sugar biomarkers. As stated also in our reply to Reviewer#1, we will gladly address this issue and other relevant issues for paleo applications more explicitly during revision.

*Several other suggestions regarding the quality of figures:*

*Figure 1. This figure is very busy. It needs to be split into 3 figures (climate chamber conditions, d2H of waters and n-alkanes, and d18O of waters and sugars). At this scale, it is difficult to see what is going on.*

→ During revision we will ensure a good scale for all figures and reorganize especially figure 1.

*Figure 2. The font in the legend is too small. I expect that I won't be legible when transferred to a published figure. Also, when showing biomarkers, it would be useful to identify the three different plants used for this study, like is done in Figure 3.*

→ Will readily be changed.

*Figure 3. The choice of symbol colours when showing different plants could be improved. At this scale, it is difficult to see the difference between purple and black. The symbols themselves could also be made larger.*

→ Will readily be changed.

*Figure 6. The same issue as mentioned for Figure 1, i.e., the figure should be split into two figures to make the text more legible. Also, does it make sense to show tank water data as a "box"? It would look better if plotted as a single data point.*

→ During revision we will ensure a good scale for all figures and reorganize especially figure 6. We will add a line for the tank water instead of a boxplot with one data point.

Literature

Hepp, J., Zech, R., Rozanski, K., Tuthorn, M., Glaser, B., Greule, M., Keppler, F., Huang, Y., Zech, W. and Zech, M.: Late Quaternary relative humidity changes from Mt. Kilimanjaro, based on a coupled $^2$H-$^{18}$O biomarker paleohygrometer approach, Quaternary International, 438, 116–130, doi:10.1016/j.quaint.2017.03.059, 2017.

Hepp, J., Wüthrich, L., Bromm, T., Bliedtner, M., Schäfer, I. K., Glaser, B., Rozanski, K., Sirocko, F., Zech, R. and Zech, M.: How dry was the Younger Dryas? Evidence from a coupled $\delta^2$H–$\delta^{18}$O biomarker paleohygrometer applied to the Gemündener Maar sediments, Western Eifel, Germany, Climate of the Past, 15, 713–733, doi:10.5194/cp-15-713-2019, 2019.

Hepp, J., Schäfer, I. K., Lanny, V., Franke, J., Bliedtner, M. and Rozanski, K.: Evaluation of bacterial glycerol dialkyl glycerol tetraether and $^2$H-$^{18}$O biomarker proxies along a central European topsoil transect, Biogeosciences, 17, 741–756, doi:10.5194/bg-17-741-2020, 2020.

Tuthorn, M., Zech, R., Ruppenthal, M., Oelmann, Y., Kahmen, A., del Valle, H. F., Eglinton, T., Rozanski, K. and Zech, M.: Coupling $\delta^2$H and $\delta^{18}$O biomarker results yields information on relative humidity and isotopic composition of precipitation - a climate transect validation study, Biogeosciences, 12, 3913–3924, doi:10.5194/bg-12-3913-2015, 2015.

Zech, M., Pedentchouk, N., Buggle, B., Leiber, K., Kalbitz, K., Markovic, S. B. and Glaser, B.: Effect of leaf litter degradation and seasonality on D/H isotope ratios of *n*-alkane biomarkers, Geochimica et Cosmochimica Acta, 75(17), 4917–4928, doi:http://dx.doi.org/10.1016/j.gca.2011.06.006, 2011.

Zech, M., Werner, R. A., Juchelka, D., Kalbitz, K., Buggle, B. and Glaser, B.: Absence of oxygen isotope fractionation/exchange of (hemi-) cellulose derived sugars during litter decomposition, Organic Geochemistry, 42(12), 1470–1475, doi:http://dx.doi.org/10.1016/j.orggeochem.2011.06.006, 2012.

Zech, M., Tuthorn, M., Detsch, F., Rozanski, K., Zech, R., Zöller, L., Zech, W. and Glaser, B.: A 220 ka terrestrial $\delta^{18}$O and deuterium excess biomarker record from an eolian permafrost paleosol sequence, NE-Siberia, Chemical Geology, 360–361, 220–230, doi:http://dx.doi.org/10.1016/j.chemgeo.2013.10.023, 2013.

---

## Author Response (AR1)

**Reply to Handling Editor Helge Niemann**

by Johannes Hepp

*Comments to the Author:*

*Dear Johannes,*

*first of all, I would like to apologies for the late reply. You should have received this message already some weeks ago, but this somehow got lost in our system. I hope that this delay did not cause problems for you or on of the co-authors.*

*Both reviewers were quite positive about your MS but raised some important issues that I fully support. Based on your answers in the discussion forum, you're seemingly on top of these points and I would thus like to invite you to submit a revised version of your MS where the reviewer's concerns are addressed.*

*Again my apologies for the delay. With the best wishes, Helge Niemann*

→ We are very grateful to the Handling Editor Helge Niemann for the invitation to submit a revised version of our manuscript. Please find attached the point-to-point replies to the Referee Comments.

---

## Author Response (AR2)

**Reply to Handling Editor Helge Niemann**

by Johannes Hepp

*Comments to the Author:*

*Dear Johannes Hepp,*

*your revisions were considered by two reviewers and both think that the MS has improved and is now acceptable for publication; a point of view I support. There are a few minor things that you should check. Some sentences are rather lengthy/complicated, so those could perhaps be made a bit easier for the reader. Also, please check spelling and grammar throughly.*

*With the best wishes, Helge Niemann*

→ We are very grateful to the Handling Editor Helge Niemann for accepting the revised version of our manuscript after technical corrections.

Best regards, Johannes Hepp